# Nodal signaling regulates asymmetric cellular behaviors, driving clockwise rotation of the heart tube in zebrafish

Hinako Kidokoro [1,2,3✉], Yukio Saijoh[3,4,5] & Gary C. Schoenwolf [3]

Clockwise rotation of the primitive heart tube, a process regulated by restricted left-sided Nodal signaling, is the first morphological manifestation of left-right asymmetry. How Nodal regulates cell behaviors to drive asymmetric morphogenesis remains poorly understood. Here, using high-resolution live imaging of zebrafish embryos, we simultaneously visualized cellular dynamics underlying early heart morphogenesis and resulting changes in tissue shape, to identify two key cell behaviors: cell rearrangement and cell shape change, which convert initially flat heart primordia into a tube through convergent extension. Interestingly, left cells were more active in these behaviors than right cells, driving more rapid convergence of the left primordium, and thereby rotating the heart tube. Loss of Nodal signaling abolished the asymmetric cell behaviors as well as the asymmetric convergence of the left and right heart primordia. Collectively, our results demonstrate that Nodal signaling regulates the magnitude of morphological changes by acting on basic cellular behaviors underlying heart tube formation, driving asymmetric deformation and rotation of the heart tube.

[1] Organization for Research Initiatives and Development, Doshisha University, Kyotanabe, Kyoto 610-0394, Japan. [2] Department of Cell Biology, National Cerebral and Cardiovascular Center Research Institute, Suita, Osaka 564-8565, Japan. [3] Department of Neurobiology, University of Utah School of Medicine, Salt Lake City, UT 84112-9458, USA. [4] Department of Nutrition and Integrative Physiology, University of Utah College of Health, Salt Lake City, UT 84112, USA. [5] Eccles Harrison Cardiovascular Research and Training Institute, University of Utah, Salt Lake City, UT 84112, USA. ✉email: hkidokor@mail.doshisha.ac.jp

The vertebrate heart originates from the bilateral anterior fields of lateral plate mesoderm (LPM). The left and right primordia fuse at the midline to form the heart tube, which is initially short and almost bilaterally symmetric[1,2]. Subsequently, the heart tube undergoes rapid elongation and directional looping, during which it is remodeled into the characteristic configuration required for efficient blood circulation. Abnormal heart looping causes abnormal connections between the heart chambers and great vessels[1,3].

Directional heart looping is regulated by the secreted TGF-β family protein Nodal, which is known to be an evolutionarily conserved left determinant in vertebrates. *Nodal* is asymmetrically expressed in the LPM, including heart precursor cells, and disruption of its normal left-sided expression results in aberrant heart looping[4–10] as well as in impaired left-right asymmetry of other visceral organs[11,12]. Transcriptomic and experimental studies have suggested numerous molecules as downstream targets of Nodal signaling in heart looping including genes involved in myocardial differentiation[4], cell proliferation[4], and remodeling of extracellular matrix[4,13,14] and actomyosin cytoskeleton[14,15]. However, how they affect cell behaviors underlying morphogenesis, driving tissue-scale asymmetry remains unknown.

Heart looping starts with a clockwise rotation of the heart tube in chick, mouse, and zebrafish embryos. In the chick, the linear heart tube turns its original ventral midline (the boundary of the left and right primordia) to the right by rotation, while rapidly elongating and bending ventrally, making a C-shaped loop[2,16–18]. Mechanical simulations suggested that rotation at the heart poles are sufficient to drive directional heart looping[19]. A recent study in mice also showed that rightward (clockwise) rotation of the arterial pole and subsequent leftward deformation of the venous pole generate the helical shape of the heart loop[20]. Conditional mouse mutants lacking *Nodal* specifically in the mesoderm, exhibit reduced or reversed rotation of the arterial heart pole[4]. Additionally, *Nodal*-expressing cells mainly contribute to the heart tube poles where they amplify asymmetries, generating the heart loop[4,20,21].

In zebrafish, the left and right heart primordia fuse at the ventral midline, forming a flattened, symmetric disc, which subsequently transforms into a cone, and then extends into a linear tube (Fig. 1)[22,23]. During the disc-to-tube transformation, the cardiac disc rotates clockwise, while the right heart primordium involutes ventrally[13,24–27]. Consequently, the heart tube extends toward the anterior-left direction, which is referred to as cardiac jogging, with the left and right primordia generating the dorsal roof and ventral floor of the heart tube, respectively. Active migration of myocardial cells is considered to drive the disc-to-tube transformation and the leftward extension of the heart tube. The zebrafish Nodal homolog *southpaw* (*spaw*) is suggested to increase the migration speed of left myocardial cells, leading to the clockwise rotation and the leftward movement of the disc[8,14,26]. Loss of Spaw signaling results in almost no rotation of the cardiac disc, and the heart tubes remain

mostly in the midline[9,15,28]. Live imaging studies have shown that myocardial cells migrate as a coherent population[13,24,26], maintaining cellular contacts between neighboring cells. But what behaviors of individual cells cause cohort migration is poorly characterized, and hence the mechanism by which Spaw increases the migration speed remains unclear. The polarized epithelial organization of myocardial cells is essential for cohort migration and morphogenesis, as disruption of cell polarity proteins results in immobilization of myocardial cells and a failure of heart tube formation and extension[24,25,29].

Here, we aimed to characterize the cellular basis of heart tube morphogenesis and the cell behaviors that are modulated by Nodal signaling. Using high-resolution imaging of living zebrafish embryos, we show that the cardiac disc transforms into a linear tube initially through oriented cell rearrangement and subsequently by cell shape changes. These cell behaviors drive convergent extension (CE) of the disc: converging the disc circumferentially, while extending it perpendicularly. Interestingly, left myocardial cells more actively rearrange and change their shapes as compared to right cells, leading to more rapid convergence of the left primordium. Loss of Nodal signaling abolished left-right asymmetric cell behavior as well as the asymmetric convergence of the left and right primordia. Collectively, our results suggest that Nodal signaling directs heart rotation by promoting cell behaviors that convert the flat heart primordia into a tube, driving asymmetric deformation of the left and right halves of the heart tube.

## Results

**The cardiac disc transforms into a linear tube by convergent extension.** We have previously visualized tissue dynamics during heart tube formation in the chick, showing that the heart tube forms through CE[30]. We speculated that the zebrafish cardiac disc might similarly undergo CE. To test this hypothesis and further elucidate cellular behavior underlying the transformation of the flat heart primordia into a tube, we performed live imaging of heart tube formation with time-lapse fluorescence confocal microscopy. For this purpose, we generated a zebrafish transgenic line *Tg(myl7:EGFP-CAAX)^ncv536Tg* that expresses membrane-targeted GFP specifically in myocardial cells under the control of the *myosin light chain 7* (*myl7*) promoter[31]. We tracked individual cells of the left primordium to examine how as a group they form a tube. We found that the cardiac disc converged circumferentially while extending perpendicularly (Fig. 2 and Supplementary Movies 1 and 2). The left heart primordium on average reduced its circumferential length by ~50% in 9 h, while extending perpendicularly by 80% (n = 3 embryos, Supplementary Fig. 1). Convergence occurred toward the anterior seam of the left and right primordia (yellow arrow in Fig. 2b, c); consequently, cells in the posterior part of the disc moved a greater distance than cells in the anterior part, as previously reported[13,26]. We further found that myocardial cells in the cardiac disc undergo cell-cell intercalation along the converging axis: they were circumferentially elongated, and intercalated preferentially along the circumferential direction (Fig. 3a–c and Supplementary Movie 3). As a result, cells initially arrayed in a row circumferentially within the cardiac disc became rearranged into two or more rows of cells (Fig. 3d and Supplementary Movie 4), which would contribute to convergence of the disc. The direction of cell intercalation is consistent with what we observed in the chick (the anteroposterior direction in the LPM)[30], suggesting that the cellular basis of heart tube morphogenesis is well conserved across species. Additionally, we observed dynamic cell shape changes of myocardial cells. Initially circumferentially elongated, they remarkably shortened their long axes as the heart tube formed (Fig. 3b, c).

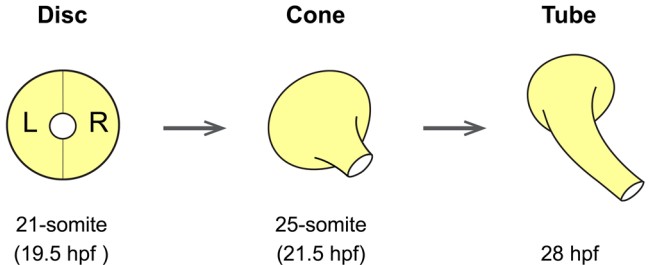

**Disc** **Cone** **Tube**

21-somite (19.5 hpf) 25-somite (21.5 hpf) 28 hpf

**Fig. 1 Schematic representation of zebrafish heart tube formation (dorsal view).** The flattened cardiac disc, which forms by fusion of the left (L) and right (R) primordia, transforms into a cone, and subsequently into a linear tube.

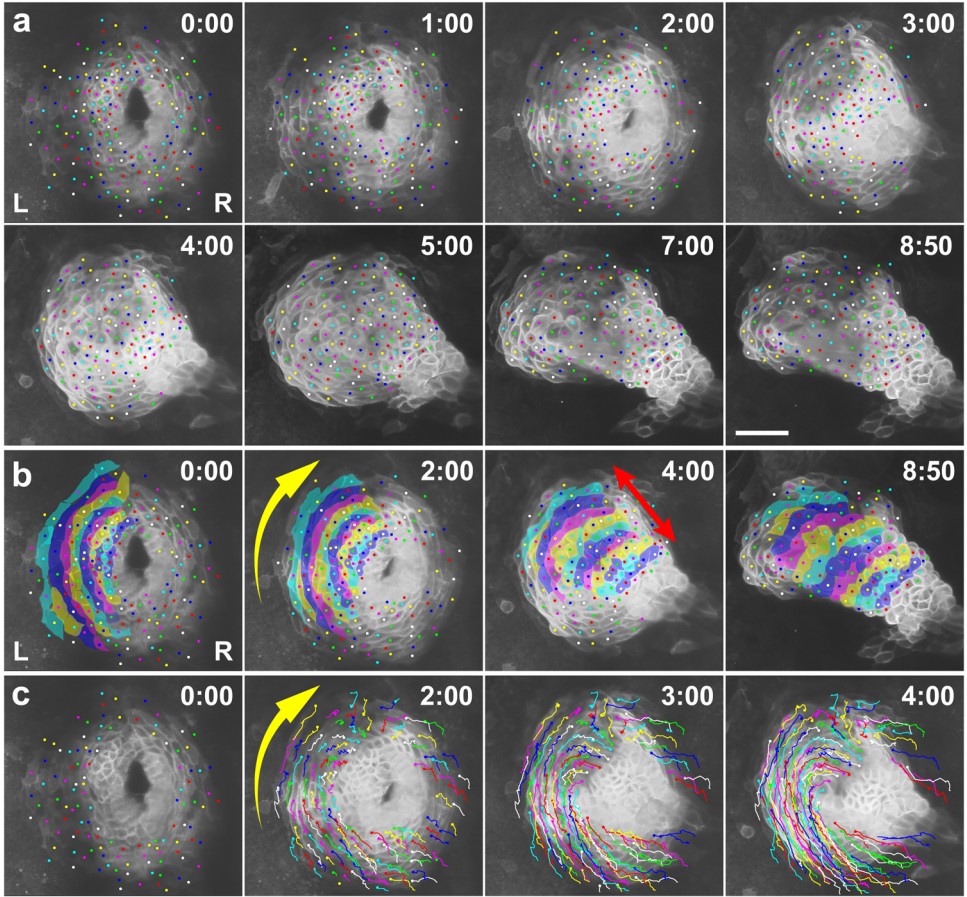

**Fig. 2 The cardiac disc undergoes convergent extension. a** Selected images from a confocal time-lapse recording of a *Tg(myl7:EGFP-CAAX)^{ncv536Tg}* embryo starting at 20 hpf (Supplementary Movie 1). Cell membranes of myocardial cells are specifically labeled with GFP in the transgenic line. Dorsal view (anterior to the top). Relative times after the initiation of the recording (h:min) are indicated in the upper right corner. Color dots indicate tracking of individual myocardial cells. Images are representative of $n \geq 10$ embryos. Scale bar = 50 µm. **b** Annotated duplicate images. Cells in the left primordium are colored. The group of cells converged in the circumferential direction toward the anterior seam of the left (L) and right (R) primordia (yellow arrow in **b**), while extending perpendicularly (red arrow in **b**) as the disc transformed into a tube. **c** Duplicated images overlaid with cell trajectories. Cell rearrangement and cell shortening occurred circumferentially toward the anterior seam of the left and right primordia (yellow arrow in **c**).

In both the chick and zebrafish, cell proliferation within the early heart tube is very low, whereas precursor cells near the heart poles display high proliferation[29,32,33]. Therefore, it is thought that cell proliferation within the tube contributes only subtly to heart tube elongation, and cell recruitment plays a more important role[34–37]. In agreement with this, in our cell tracking, we observed little cell division and cell death within the heart primordia throughout the disc-to-tube transition (Fig. 2 and Supplementary Movies 1 and 2, note that cell numbers within the colored cell groups barely change between the starting and ending time points). It was previously shown that the zebrafish heart tube at 24 hpf contains ~150 cardiomyocytes. In our cell tracking, the cardiac disc at 20 hpf already contained ~150 cardiomyocytes ($n = 3$). In the next 10 h, 30–50 newly GFP-expressing cells incorporated into the periphery of the disc (venous pole) and the arterial pole. These observations, together with previous reports, suggest that the cardiac disc transforms into a tube through rearrangement and shape change of constituent cells, and incorporation of newly-differentiating cells to heart poles also modestly extends the heart tube.

**Convergence of the cardiac disc is initially driven by oriented cell rearrangement and subsequently by cell shape change during heart tube formation**. We quantified at an individual cell

level how changes in cell arrangement and cell shape accounted for the convergence of the cardiac disc. For this purpose, we analyzed cells circumferentially arrayed in a row within the disc as a unit, as shown in Fig. 4b (different colors for each cell array; cells in the left primordium were analyzed). We measured the entire length of each cell array (black line in Fig. 4c and Supplementary Movie 4) and the length (long axis) of constituent cells (blue lines in Fig. 4d and Supplementary Movie 4) over time to examine how shortening of individual cells accounted for convergence at the tissue level. To quantify how much each cell array reduced its length by cell rearrangement, we measured the overlap between the neighboring cells within an array (red lines in Fig. 4d), considering the overlap as a loss of cell length. This analysis revealed that cell rearrangement and cell shape change drove convergence of the heart primordia at different time periods of heart tube formation (Fig. 4e and Supplementary Fig. 2). In the early phase of heart tube formation (see plots from time 0 to 3 h in Fig. 4e1), the length reduction of cell arrays (black plots in Fig. 4e) were highly correlated with the length loss caused by cell overlaps (red plots in Fig. 4e), suggesting that convergence of the cardiac disc at the early stage is mainly driven by cell rearrangement. During this early period, cells stayed elongated or slightly increased their length (blue plots in Fig. 4e1). In contrast, in the later phase (see plots from 3 h onward in Fig. 4e1), cells progressively reduced their length. In concordance with this cell

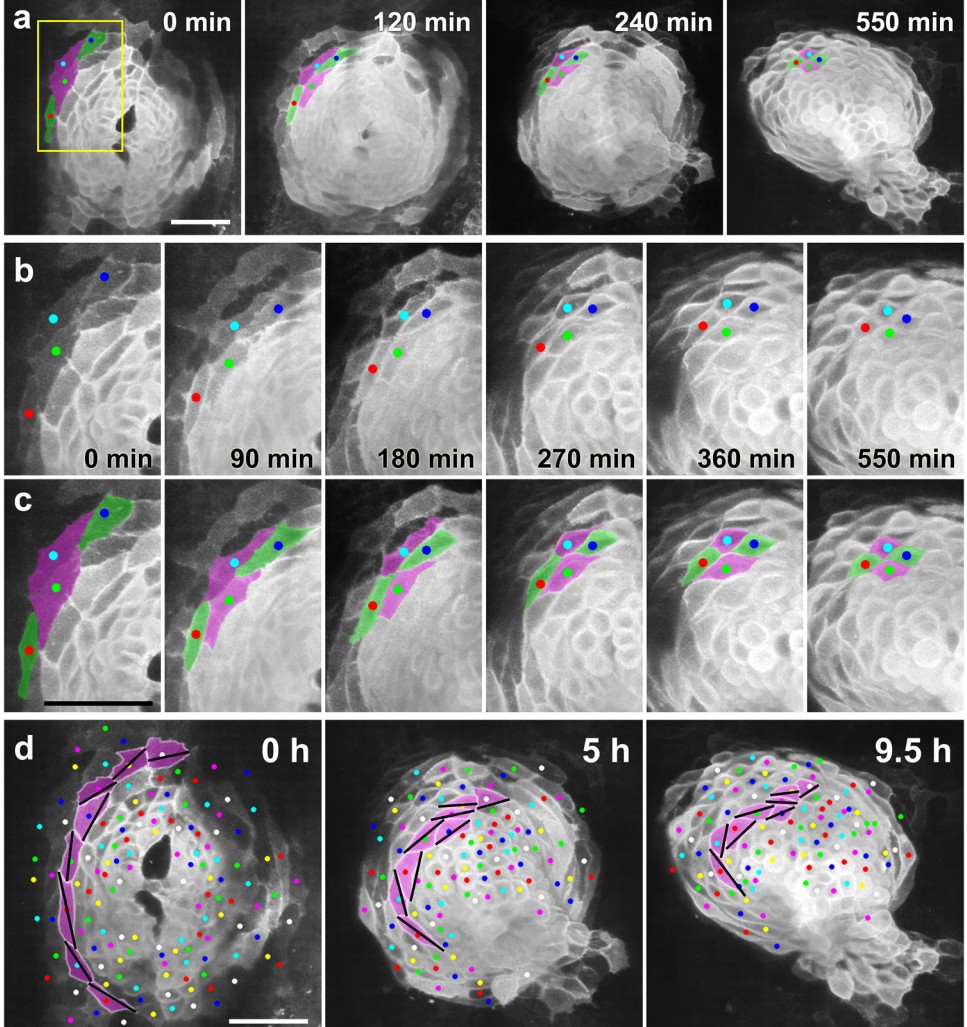

**Fig. 3 Myocardial cells undergo cell-cell intercalation and cell shortening along the circumferential direction during heart tube formation. a** Selected images from a confocal time-lapse recording of a *Tg(myl7:EGFP-CAAX)*^ncv536Tg^ embryo starting at 19–20 hpf (Supplementary Movie 3), showing intercalating myocardial cells (green). Dorsal view (anterior to the top). Scale bar = 50 μm. **b** Enlargement of the region indicated by the box in **a**. **c** Colored, corresponding images to those in **b**. Initially the two separated cells (green) intercalated between their neighbors (magenta), coming into contact with one another and thereby separating the two neighboring cells. Note that myocardial cells were circumferentially elongated at the beginning (0 min), but progressively shortened into a round shape (550 min). Scale bar = 50 μm. **d** An array of cells in the left heart primordium are colored with magenta. Bars indicate the long axes of the cells. As a result of the oriented cell rearrangement, cells initially arrayed in a single row in the cardiac disc piled up to generate two or more rows. Cell intercalation was consistently observed in n ≥ 10 embryos. Scale bar = 50 μm.

length reduction, the length of cell arrays was also reduced, suggesting that the later convergence of the cardiac disc is mainly driven by cell shortening. During the later period, the overlap between cells barely changed or slightly decreased. Of note, throughout heart tube formation, myocardial cells oriented their long axes nearly along the circumference of the heart disc/tube (Supplementary Movie 4), suggesting that they are planar polarized. Such cell polarization is considered to be important for efficient tissue convergence through cell rearrangement and cell shortening. In concordance with this notion, the sum (magenta plots) of the relative changes of cell length (blue plots) and the length loss caused by the cell overlap (red plots) closely matched to the length change of cell arrays (black plots in Fig. 4e), suggesting that convergence of the heart primordia is caused almost entirely by cell rearrangement and cell shortening. Collectively, these data suggest that convergence of the heart primordia is initially driven by cell rearrangement and subsequently by cell shape change. The trend that cell rearrangement occurred first,

followed by cell shortening, was consistent across the peripheral-to-inner cell arrays (compare Fig. 4e2 with 4e3). However, cell shortening started earlier in the inner region (Fig. 4e3) than in the peripheral region (Fig. 4e2), and hence a large portion of the entire convergence of inner cell arrays was caused by it (Fig. 4e3). This may be because the heart tube forms progressively from the central to peripheral sides of the cardiac disc. That is, inner cells may be developmentally more advanced than peripheral cells (note that GFP expression induced by the *myl7* promoter appears earlier in inner cells in Fig. 4a). In summary (Fig. 4f), in the early phase of heart tube formation, myocardial cells become circumferentially elongated and actively rearrange to converge the cardiac disc, forming a tube. Subsequently, cells stop exchanging neighbors, and start shortening their long axes to become rounded, thereby further narrowing the heart tube.

**The left primordium more rapidly converges than the right primordium during rotation of the cardiac disc.** The cardiac

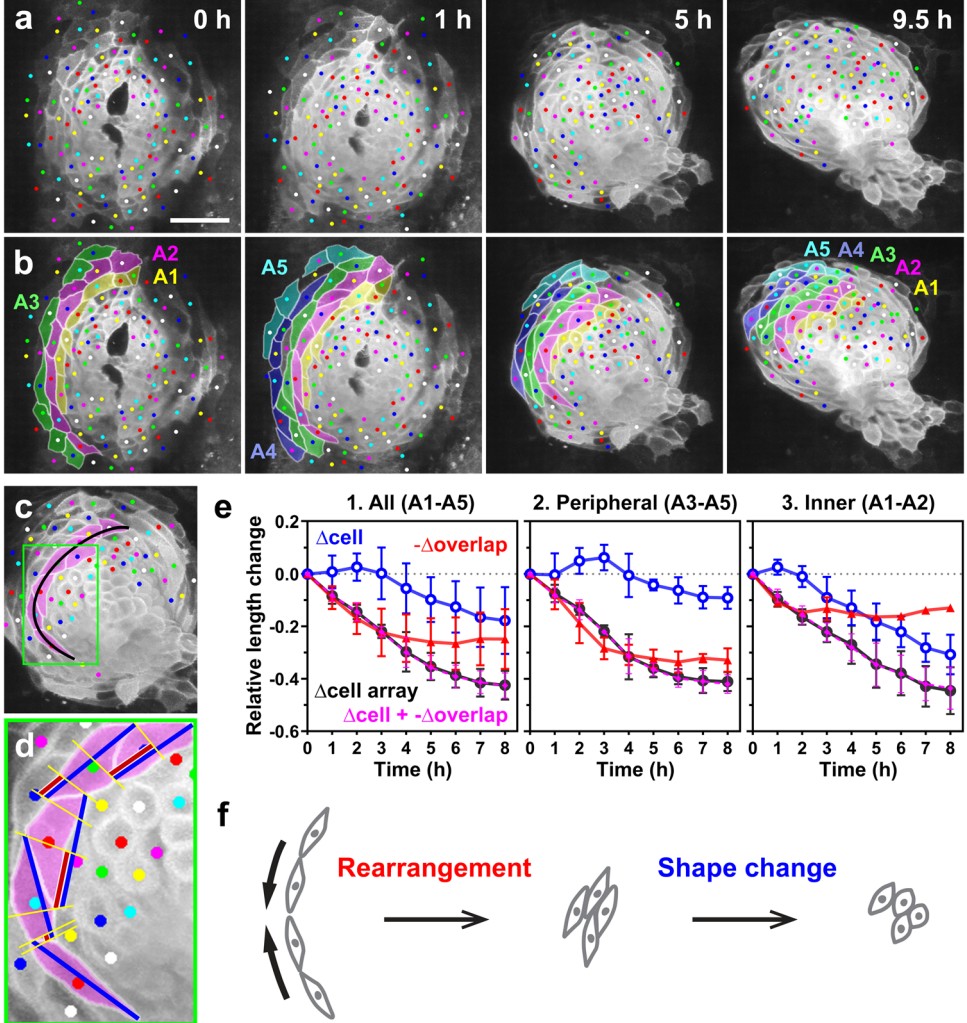

**Fig. 4 Oriented cell rearrangement and cell shape change drive convergence of the cardiac disc at the early and late phases of heart tube formation, respectively. a**, **b** For quantitative analysis, we considered cells circumferentially arrayed in a row (A1–A5, each cell array is shown in a different color in **b**) as a unit. **a** Selected images from a confocal time-lapse recording of a *Tg(myl7:EGFP-CAAX)*$^{ncv536Tg}$ embryo starting at 19–20 hpf. Dorsal view (anterior to the top). Scale bar = 50 μm. **b** Colored, corresponding images to those in **a**. Cells in the left primordium (colored cells) were analyzed. Cell arrays are numbered from the inner to the peripheral order (A1–A5). We measured the length of each cell array (black line in **c**), the length of constituent cells (blue lines in **d**) and the overlap length between neighboring cells (red lines in **d**) every 1 h. **c** A cell array is indicated by magenta. **d** Enlargement of the region indicated by the box in **c**. **e** Relative length changes of cell arrays (black), cells (blue), and loss of the cell lengths caused by the cell overlap (red) in five cell arrays (**e**1, A1–A5 indicated in **b**), in the peripheral cell arrays (**e**2, A3–A5) and in the inner cell arrays (**e**3, A1–A2). Magenta plots represent the sum of relative changes of cell length (blue) and the length loss by the cell overlap (red). *n* = 5 cell arrays, 4–8 cells per each cell array (1 embryo). Data are normalized relative to sum of cell lengths at the initial time point to eliminate the effects of each cell array size. Means ± s.d. are shown. Corresponding graphs indicating individual data points are shown in Supplementary Fig. 2. **f** Model for convergence of the cardiac disc based on our quantitative analysis. Circumferential convergence is initially driven by oriented cell rearrangement and subsequently by cell shortening.

disc is almost bilaterally symmetric just after the left and right primordia merge at the midline. Our cell tracking showed that at this stage left and right heart primordia equally compose each half of the cardiac disc (Fig. 5a, 1 h). At this point, the anterior and posterior borders of the left and right primordia were at 12 and 6 o'clock, respectively. However, as the heart tube formed, the posterior border shifted to 7 o'clock (Fig. 5a, 5 h), and then to 9 o'clock (Fig. 5a, 9.5 h, *n* = 8/8 embryos), suggesting that the left primordium converged more rapidly than the right one. We quantified the circumferential length of the peripheries of the left and right primordia (Supplementary Fig. 3b, c): the ratio of the left peripheral length to the right one was initially 1:1 ± 0.04 (s.d.) at 0 h, subsequently became 0.67:1 ± 0.04 (s.d.) at 4 h, and 0.5:1 ± 0.04 (s.d.) at 8.5 h (*n* = 3 embryos, Supplementary Fig. 3c, uninjected embryos), supporting our notion. We quantified the

cell array convergence similarly to Fig. 4. Because the right primordium involutes ventrally and becomes hidden as the heart tube forms, we only could quantify cells in the anterior portion of the primordium (colored regions in Fig. 5b and Supplementary Movie 5) in a more limited time window as compared with that in Fig. 4. The results showed that both the left and right heart primordia undergo circumferential convergence, but the convergence speed is asymmetric between the left and right halves (Fig. 5c and Supplementary Fig. 4a). The peripheral region of the left primordium converged more rapidly than the corresponding region of the right primordium (*n* = 3 embryos, Fig. 5c1), whereas no significant left-right difference was found in the inner region (Fig. 5c2).

We next asked what cellular behavior is responsible for the asymmetric convergence of the left and right heart primordia.

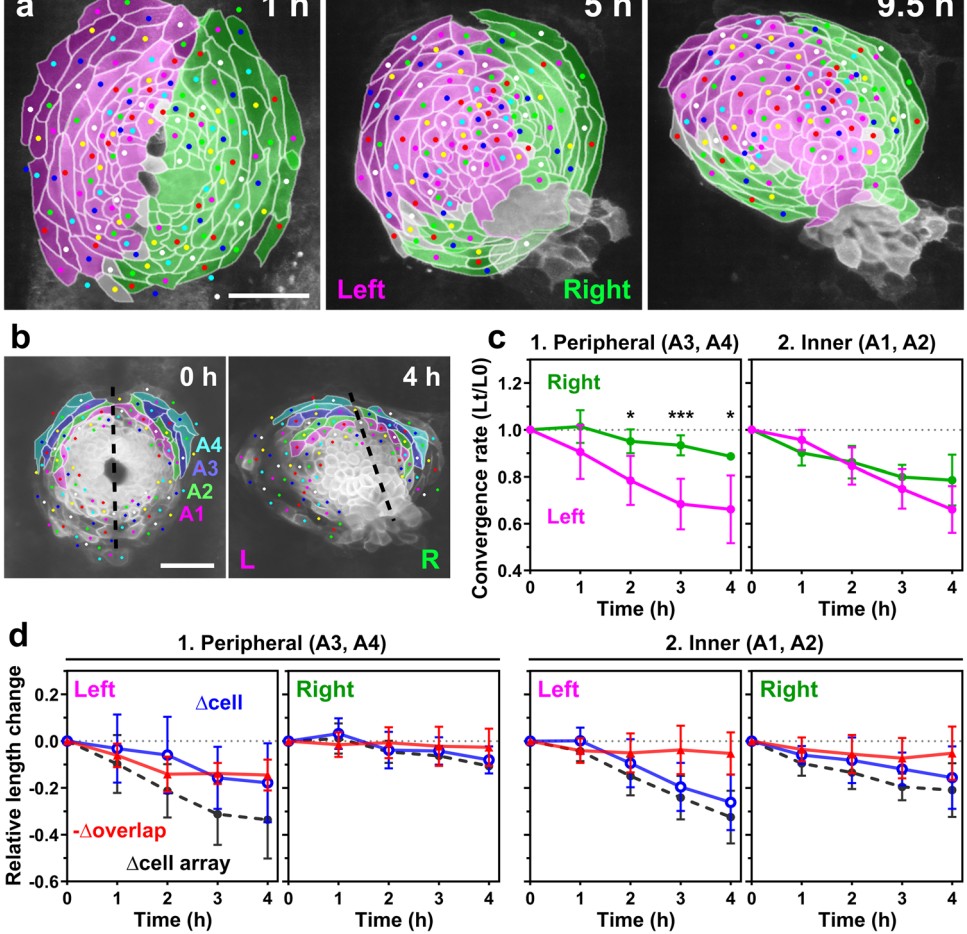

**Fig. 5 The left and right heart primordia undergo asymmetric convergence during heart tube rotation. a** Cells derived from the left (magenta) and right (green) primordia are color-coded. The posterior border of the left and right primordia shifted from 6 o'clock to 7–9 o'clock, suggesting that the left primordium converged more rapidly than the right one. Cells with ambiguous left-right origins are left uncolored. The original corresponding images are shown in Fig. 4a. Scale bar = 50 μm. **b**–**d** Quantification of the convergence of the left (L) and right (R) primordia. We analyzed the anterior portion of the primordia (colored regions in **b**) because the posterior portion of the right primordium became hidden due to their involution shortly after heart rotation started. The dashed lines in **b** indicate the boundary of the left and right primordia. Scale bar = 50 μm. **c** Plots of the relative lengths (Lt/L0) of peripheral (1) and inner (2) cell arrays in wild-type embryos. Lt is the length at time t, L0 is the length at t = 0. The left (magenta plots) peripheral cell arrays (A3 and A4 in **b**) converged more rapidly than the right (green plots) ones, whereas no significant difference was observed between the left and right cell arrays in the inner region (A1 and A2 in **b**). *$p < 0.05$, **$p < 0.01$, ***$p < 0.005$ (two-tailed $t$-test assuming unequal variances, $n = 3$ embryos). Means ± s.d. are shown. **d** Relative length changes of cell arrays (black), cells (blue), and loss of the cell lengths caused by the cell overlap (red) in the peripheral (1) and inner (2) cell arrays in wild-type embryos. $n = 3$ embryos. Means ± s.d. are shown. For **c**, **d**, corresponding graphs indicating individual data points are shown in Supplementary Fig. 4.

Quantification revealed that left myocardial cells are more active both in cell rearrangement and cell shortening than right cells (Fig. 5d and Supplementary Fig. 4b). Similar to results shown in Fig. 4e, cell rearrangement accounted for a large portion of the convergence of peripheral cell arrays at the early stage, whereas cell shortening accounted for most of it at the later stage (Fig. 5d1). The convergence of the inner cell arrays was driven mostly by cell length reduction, and only slightly by cell rearrangement (Fig. 5d2), as compared with results shown in Fig. 4e. This probably reflects the difference in developmental stages analyzed. For the left-right comparison, measurements only could be performed when both left and right cells in similar relative positions express sufficient levels of GFP. This limitation often delayed the initial time points of the measurements as compared to that in Fig. 4e. Importantly, in both the left and right primordia, the long axes of myocardial cells were circumferentially aligned during heart tube formation, suggesting that that

cell polarization is not specific for left cells, but rather occurs in the entire population of myocardium.

**Left-right asymmetric convergence is due to intrinsic differences between the left and right cells.** Is the asymmetric convergence of the left and right heart primordia due to distinct intrinsic properties of the left and right cells? As the cardiac disc rotates clockwise, it is likely that a different quantity and quality of mechanical stress is loaded on the left and right heart primordia, which may affect behaviors of left and right cells. To remove such a mechanical bias and see whether left and right cells still exhibit different behaviors in the absence of unequal mechanical conditions, we prevented fusion of the left and right heart primordia, generating two heart tubes that develop independently. An additional advantage of this experiment is that posterior cells in the cardiac disc also could be included in the analysis. Heart fusion depends on the adjacent endoderm, whose

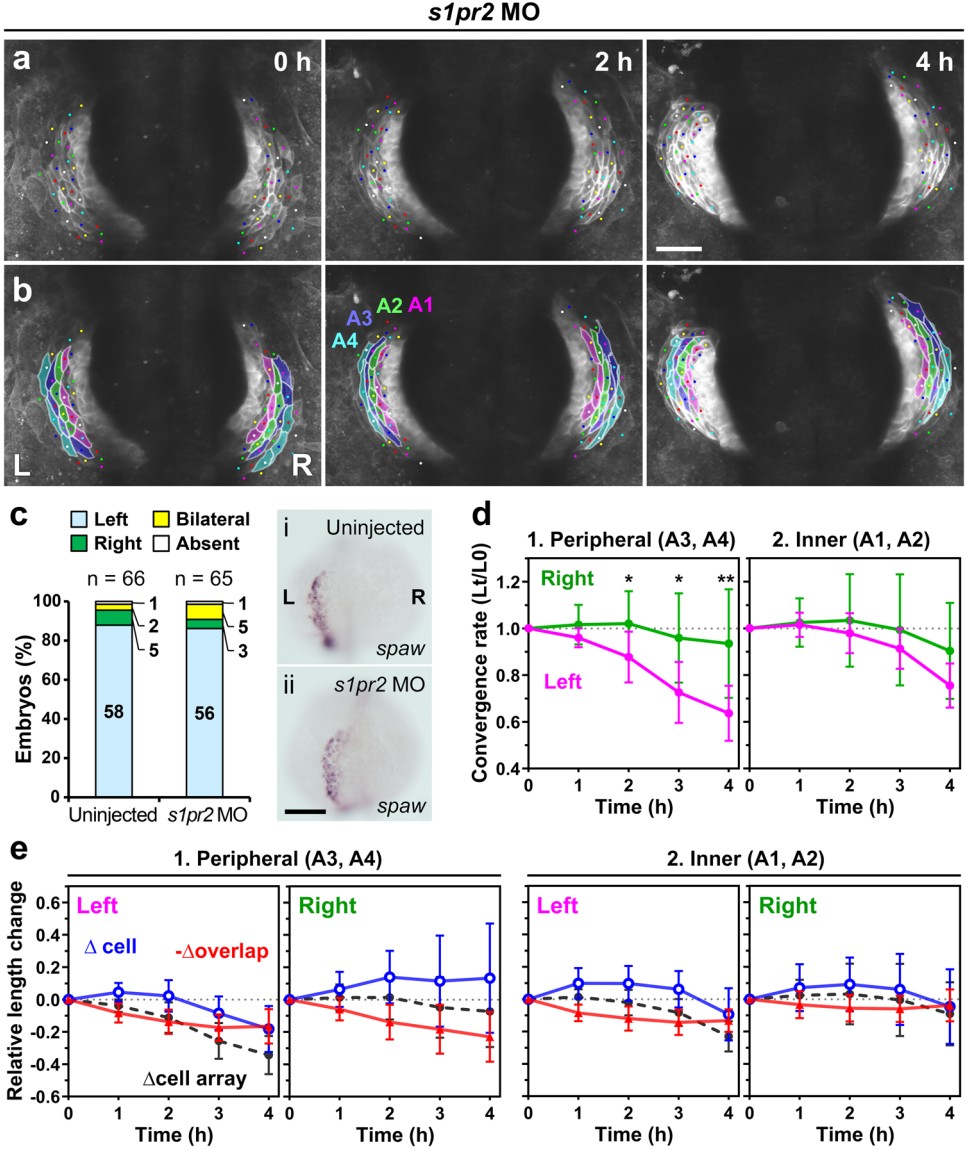

**Fig. 6 Left-right asymmetric convergence occurs independently of mechanical bias caused by heart rotation. a, b** Embryos with cardia bifida were created by injecting *s1pr2* MO, which inhibits endoderm convergence and subsequent heart primordia fusion. Asymmetric convergence still occurred in the two hearts. **a** Selected images from a confocal time-lapse recording of a *s1pr2* MO-injected *Tg(myl7:EGFP-CAAX)*[ncv536Tg] embryo starting at 19–20 hpf (Supplementary Movie 6). Dorsal view (anterior to the top). Scale bar = 50 μm. **b** Colored, corresponding images to those in **a**. Four cell arrays are labeled with different colors (A1–A4). **c** *spaw* mRNA expression in uninjected (i) and *s1pr2* MO-injected (ii) embryos was assessed by in situ hybridization. The bar graph shows percentage of control (uninjected siblings, *n* = 66) and *s1pr2* MO-injected embryos (*n* = 65) with left-sided, right-sided, bilateral, or absence of *spaw* expression in the lateral plate mesoderm (LPM). Morpholino injection did not perturb the normal left-sided *spaw* expression in the LPM. Scale bar = 200 μm. **d** Plots of the relative lengths (Lt/L0) of peripheral (1) and inner (2) cell arrays in *s1pr2* MO-injected embryos (cardia bifida). The left cell arrays in the peripheral region (A3 and A4) converged more rapidly than the right ones, whereas no significant difference was observed between the left and right cell arrays in the inner region (A1 and A2), similar to that as in normal embryos shown in Fig. 5. *$p < 0.05$, **$p < 0.01$, ***$p < 0.005$ (two-tailed *t*-test assuming unequal variances, *n* = 4 embryos). Means ± s.d. are shown. **e** Relative length changes of cell arrays (black), cells (blue), and loss of the cell lengths caused by the cell overlap (red) in the peripheral (1) and inner (2) cell arrays in *s1pr2* MO-injected embryos. *n* = 4 embryos. Means ± s.d. are shown. For **d, e**, corresponding graphs indicating individual data points are shown in Supplementary Fig. 5.

convergent movement pulls the paired heart primordia toward the midline where they fuse to form the heart tube[38,39]. S1pr2 functions within the endoderm to regulate endodermal convergence, and when its function is defective the bilateral heart primordia fail to fuse, resulting in cardia bifida, but otherwise the embryos develop normally including myocardial differentiation[38,40–42]. We injected *s1pr2* morpholino oligonucleotides (MO)[38,40,43] into the embryos of *Tg(myl7:EGFP-CAAX)*[ncv536Tg] at the one- to two-cell stage. In these morphants, the left and right heart primordia developed separately as

previously shown (Fig. 6a, b)[38,40,43]. mRNA in situ hybridization confirmed that the normal left-sided expression of *spaw* in the LPM was unaltered by the *s1pr2* morpholino injection (Fig. 6c, 86%, *n* = 65). Similar to normal hearts, in cardia bifida, the left cell arrays in the peripheral region of the cardiac disc converged more rapidly than the right ones (*n* = 4 embryos, Fig. 6a, b, d1, Supplementary Fig. 5a1, and Supplementary Movie 6), whereas there was no significant left-right difference in convergence of cell arrays in the inner region (Fig. 6a, b, d2 and Supplementary Fig. 5a2). The rapid convergence of the left cell arrays was

accounted for mainly by cell rearrangement in the early phase and by cell shortening in the later phase (Fig. 6e1 and Supplementary Fig. 5b1), as observed in the normal embryos (Fig. 5d1). Additionally, both left and right cells were polarized in the circumferential direction, similarly to normal hearts. Compared to normal hearts, cell shortening of left inner cells in non-fused hearts was delayed and they also showed an increased cell length (Fig. 6e2 and Supplementary Fig. 5b2). Similarly, right cells in non-fused hearts showed an increased cell length (Fig. 6e1, e2), differing from the normal hearts where right cells barely changed their length (Fig. 5d). As the right cells in cardia bifida became more elongated, the overlap between the neighboring cells increased. However, the length of the entire cell arrays stayed the same. Collectively, in the separately developed hearts, the left primordium still exhibited more rapid convergence through more effective cell rearrangement and rapid cell shortening, suggesting that these different behaviors between left and right cells are attributed to their intrinsic distinct properties. Compared to normal hearts, cell shortening in both the left and right primordia of non-fused hearts were delayed. This may be due to the presence of defective endoderm or altered physical conditions (see "Discussion").

**Nodal signaling regulates left-right differences in cellular behavior**. We asked, using morpholino knockdown of *spaw*, whether left-specific Nodal signaling is responsible for the more rapid convergence of the left heart primordium. We first injected *spaw* morpholino[15,24,28] alone into *Tg(myl7:EGFP-CAAX)*[ncv536Tg] embryos; *spaw* expression in the LPM was completely abolished, whereas it remained normally expressed in the tail bud (100%, $n = 60$, Supplementary Fig. 3a). This was expected as Spaw itself is required to propagate *spaw* expression in the LPM via positive-feedback loops[15,28,44,45]. In these morphants, the shift of the posterior border of the left and right primordia toward 7–9 o'clock did not occur ($n = 7/9$ *spaw* MO-injected embryos, Fig. 7a, b, compare with Fig. 5a, and Supplementary Movie 7), and the heart tube formed at the midline with no rotation as previously shown[8,14,15,26]. In contrast to uninjected embryos, the peripheral lengths of the left and right primordia in *spaw* morphants were similar throughout heart tube formation (Supplementary Fig. 3c). In concordance with this, the more rapid convergence of the left peripheral region of the disc than the right corresponding region, through more active cell rearrangement and cell shortening, was abolished in these morphants, leading to loss of left-right asymmetry in the cardiac convergence ($n = 3$ embryos, Fig. 7c–e and Supplementary Fig. 6). Comparison between uninjected and *spaw* MO-injected embryos revealed that Spaw knockdown significantly reduced the convergence of the left primordium in its peripheral region (Supplementary Figs. 7–9), whereas it had no significant effect on the inner region of the left primordium and the entire right primordium. The venous end of the heart tube in *spaw* morphants was widened ($n = 6$, W4 position in Supplementary Fig. 3d–f), whereas the width in the arterial half of the heart tube was not significantly different from that of normal hearts ($n = 6$, W1–3 positions in Supplementary Fig. 3d–f). Additionally, the heart tube length of *spaw* morphants ($n = 6$) was shorter than that of uninjected embryos ($n = 6$, Supplementary Fig. 3d, e, g). These results suggest that Spaw has a striking effect on increasing the extent of convergence in the peripheral region of the heart primordium. However, in this condition, we cannot exclude the possibility that the rapid convergence of the left primordium was blocked in *spaw* morphants as a consequence of it being stuck in the midline. To remove such a mechanical constraint and to analyze the behavior of left and right cells under equivalent conditions, we next co-injected *s1pr2*

morpholino to generate cardia bifida, in conjunction with *spaw* morpholino into *Tg(myl7:EGFP-CAAX)*[ncv536Tg] embryos. In the non-fused hearts, Spaw knockdown still abolished the more rapid convergence of the left heart primordium ($n = 3$ embryos, Fig. 8a–d, Supplementary Fig. 10a, and Supplementary Movie 8), as well as the underlying cell rearrangement and cell shortening (Fig. 8e and Supplementary Fig. 10b). Consequently, there was no significant difference in the convergence speed between the left and right primordia in these morphants. As observed in embryos injected with *s1pr2* morpholino alone (Fig. 6), both the left and right cells showed a marked increase in their length, accompanied by a marked increase of the overlap between neighboring cells (Fig. 8e). These results suggest that Spaw increases the convergence speed of the left primordium by promoting cell rearrangement and rapid cell shortening.

## Discussion
Using high-resolution live imaging, we have simultaneously documented cellular and tissue dynamics, which lead to formation and rotation of the heart tube. Our results identified two key cell behaviors underlying heart tube morphogenesis: oriented cell rearrangement and cell shape change. We further revealed that Nodal signaling on the left side promotes these cell behaviors, driving asymmetric shape changes of the heart primordia. Collectively, our analyses demonstrated that the role of Nodal signaling is not to evoke different types of morphogenesis, but to modulate the magnitude of morphological changes, generating a small, but critical asymmetry in heart formation[4].

We have shown that myocardial cells undergo oriented rearrangement and shape change to rapidly remodel the cardiac disc into a tube. Both cell behaviors occurred circumferentially toward the anterior seam of the left and right primordia, which caused a longer translocation of cells originating from the posterior region of the disc anteriorly than cells from the anterior region, as observed previously[13,26]. Moreover, the left primordium converged more rapidly than the right primordium, through more active cell rearrangement and cell shortening. Previous studies have shown that myocardial cells exhibit migratory behavior during the disc-to-tube transition, with different speeds among regions within the cardiac disc: posterior and left cells move faster than anterior and right cells, respectively[8,13,14,26]. Our data suggest that this directional migration of the left/posterior cells with a higher speed might be at least partially driven by their active rearrangement and cell shortening toward the anterior direction. The previous observation that myocardial cells move as a coherent population, maintaining cell-cell contacts[13,24,26] supports this notion.

The left-right differences in cell behavior and tissue convergence were also observed in experimentally induced cardia bifida, suggesting that they occur not as consequences of heart rotation, but due to intrinsic properties of left and right cells. Loss of Nodal signaling abolished both asymmetries in cellular and tissue dynamics, and also heart rotation. Collectively, our results suggest that Nodal signaling directs heart rotation by promoting cell rearrangement and cell shape changes, leading to asymmetric convergence of the heart primordia (Fig. 9). To be clear, CE through cell rearrangement and cell shortening occurred in both the left and right primordia (Figs. 5c and 6d). Spaw knockdown only reduced the convergence of the peripheral region of the left primordium (Fig. 7d and Supplementary Fig. 7). These results suggest that the peripheral cells have basal levels of cell rearrangement and cell shape change, and that the role of Nodal is to locally enhance these levels.

How Nodal signaling promotes cell rearrangement and cell shape change remains an open question. Considering that both

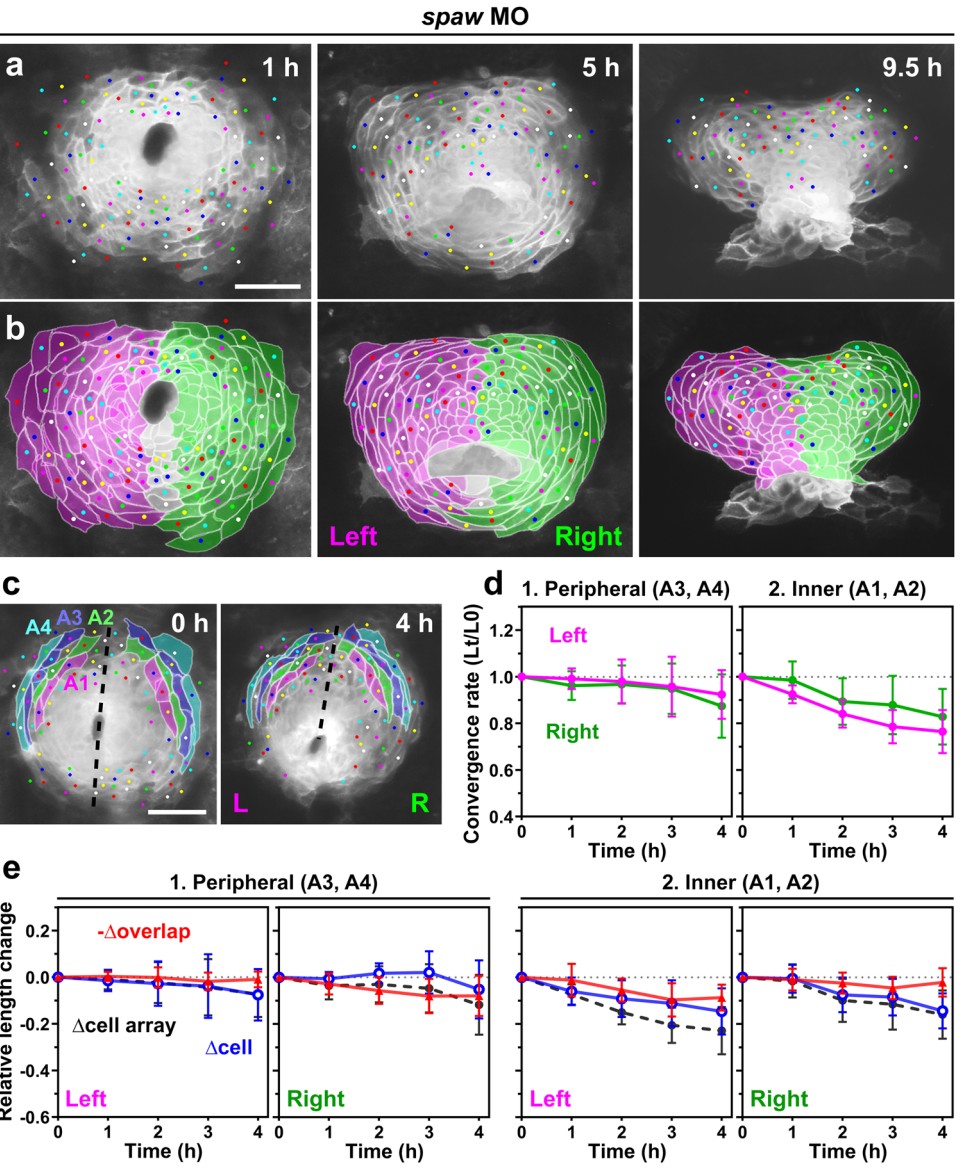

**Fig. 7 Loss of Nodal signaling abolished asymmetric convergence of the left and right heart primordia. a, b** Spaw knockdown resulted in failure of the clockwise shift of the posterior border between the left and right heart primordia. **a** Selected images from a confocal time-lapse recording of a *spaw* MO-injected *Tg(myl7:EGFP-CAAX)*^ncv536Tg embryo starting at 19–20 hpf (Supplementary Movie 7). Dorsal view (anterior to the top). Scale bar = 50 µm. **b** Colored, corresponding images to those in **a**. Cells derived from the left (magenta) and right (green) primordia are color-coded. Cells with ambiguous left-right origins are left uncolored. **c–e** Quantification of the convergence of the left (L) and right (R) primordia. The anterior portion of the primordia (colored regions in **c**) were analyzed. Scale bar in **c** = 50 µm. **d** Plots of the relative lengths (Lt/L0) of peripheral (1) and inner (2) cell arrays in *spaw* MO-injected embryos. No significant difference was observed between the left (magenta plots) and right (green plots) cell arrays in both the peripheral and inner regions (two-tailed *t*-test assuming unequal variances, *n* = 3 embryos), unlike uninjected embryos (Fig. 5). Means ± s.d. are shown. **e** Relative length changes of cell arrays (black), cells (blue), and loss of the cell lengths caused by the cell overlap (red) in the peripheral (1) and inner (2) cell arrays in *spaw* MO-injected embryos. *n* = 3 embryos. Means ± s.d. are shown. For **d**, **e**, corresponding graphs indicating individual data points are shown in Supplementary Fig. 6.

cell behaviors are achieved by regulating cell adhesion, cell polarity, and actomyosin dynamics[46–48], molecules regulating this machinery are possible candidates for downstream targets of Nodal signaling. In fact, compared to left cells, right cells appeared to maintain tight cell-cell contacts, as changes in positional relationship between neighboring cells were modest in our imaging. This observation suggests that adhesion molecules such as N-cadherin, which is expressed in the early heart and is essential for its morphogenesis[49–51], might be downregulated in left cells. Alternatively, upregulating actin/myosin dynamics also would positively regulate these cell behaviors. Nodal signaling is known to increase actin dynamics and endodermal cell motility

via Rac1 during gastrulation[52]. However, it was shown instead that both expression and phosphorylation of nonmuscle myosin are downregulated on the left side of the cardiac disc[14]. Our live imaging showed that myocardial cells are planar polarized, with elongation and intercalation occurring along the circumferential direction. Circumferential cell polarization was observed in both the left and right cells in normal and non-fused hearts, and also in hearts with *spaw* knockdown, suggesting that planar cell polarization is a fundamental mechanism underlying heart tube morphogenesis. Planar cell polarity (PCP) signaling regulates cell rearrangement during the cardiac chamber remodeling[53] and the deployment of second heart field cells[54–56]. Thus, PCP signaling

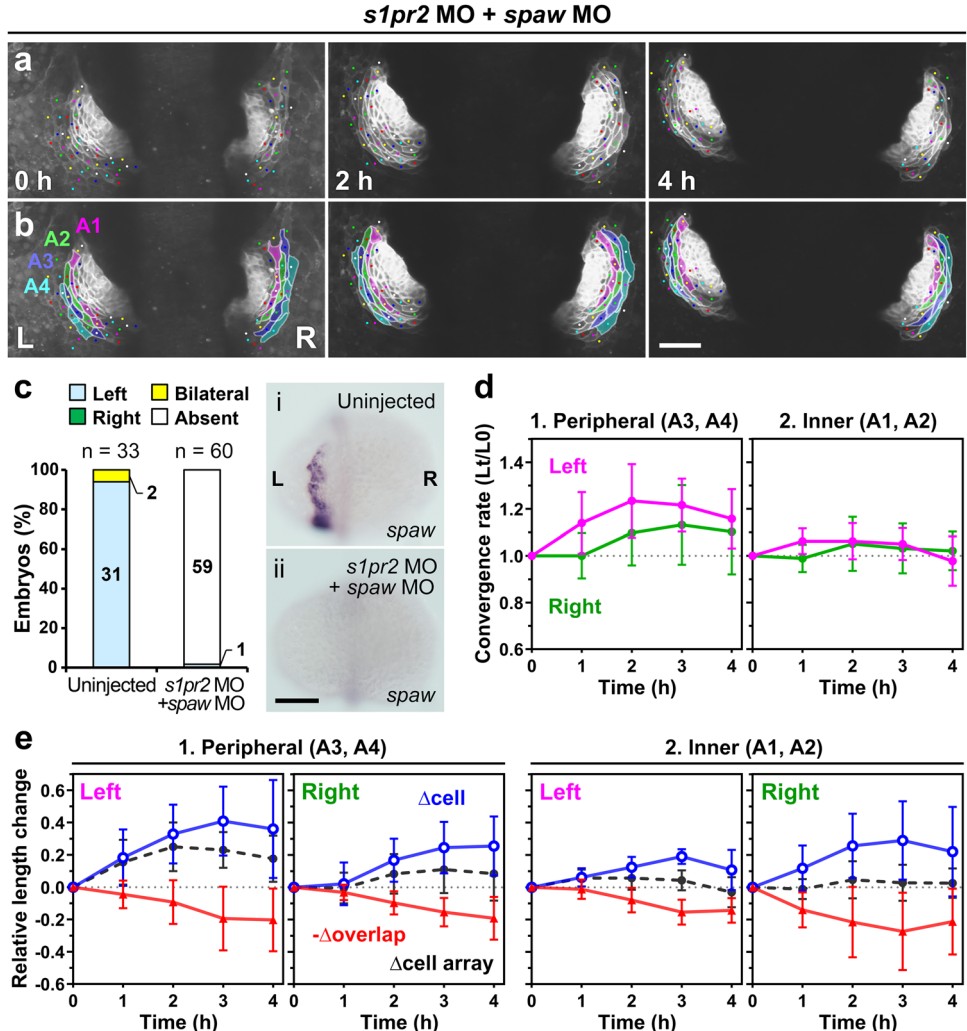

**Fig. 8 Nodal signaling regulates asymmetric convergence of the left and right heart primordia. a, b** Loss of Nodal signaling abolished asymmetric convergence of the left (L) and right (R) primordia in cardia bifida. **a** Selected images from a confocal time-lapse recording of a $Tg(myl7:EGFP-CAAX)^{ncv536Tg}$ embryo co-injected with *spaw* MO and *s1pr2* MO starting at 19–20 hpf (Supplementary Movie 8). Ventral view (anterior to the top). The images were horizontally flipped to be presented in a consistent left-right orientation with other images. **b** Colored, corresponding images to those in **a**. Four cell arrays are labeled with different colors (A1–A4). Scale bar = 50 μm. **c** *spaw* mRNA expression in uninjected (i) and MO-injected (ii) embryos was assessed by in situ hybridization. The bar graph shows percentage of control (uninjected siblings, $n = 33$), and *spaw* MO and *s1pr2* MO co-injected embryos ($n = 60$) with left-sided, right-sided, bilateral, or absence of *spaw* expression in the lateral plate mesoderm (LPM). Morpholino injection abolished *spaw* expression in the LPM. Scale bar = 200 μm. **d** Plots of the relative lengths (Lt/L0) of peripheral (1) and inner (2) cell arrays in *spaw* and *s1pr2* double morphants. In embryos injected with both *spaw* MO and *s1pr2* MO, left myocardial cell arrays did not exhibit the more rapid convergence unlike morphants injected with only *s1pr2* MO shown in Fig. 6. No significant left-right difference was found in both the peripheral and inner cell arrays (two-tailed *t*-test assuming unequal variances, $n = 3$ embryos). Means ± s.d. are shown. **e** Relative length changes of cell arrays (black), cells (blue), and loss of the cell lengths caused by the cell overlap (red) in the peripheral (1) and inner (2) cell arrays in *spaw* and *s1pr2* double morphants. Cell length in the morphants was markedly increased in both left and right cells. $n = 3$ embryos. In one out of these three embryos, only A2–A4 cell arrays were analyzed because accurate measuring was difficult in A1 cell arrays of this embryo. Means ± s.d. are shown. For **d**, **e**, corresponding graphs indicating individual data points are shown in Supplementary Fig. 10.

may similarly regulate oriented cell rearrangement/cell shortening during heart tube formation. A recent study suggested that during gastrulation, Nodal signaling promotes CE of the neuroectoderm, cell-autonomously via planar cell polarization both upstream and independent of PCP signaling[57]. Thus, it may be interesting to investigate whether a similar mechanism operates in heart rotation to enhance CE in the left primordium.

Our data showed clear left-right differences in tissue convergence in the peripheral region of the cardiac disc, but not in the inner region. Similarly, significant differences in tissue convergence between normal and Spaw knockdown hearts were found only in the peripheral region. This may reflect the localized expression site of *Nodal*. It has been shown in mice that *Nodal*-

expressing cells barely contribute to the initially formed ventricles, instead mainly contributing to the heart poles, which are generated by cell populations incorporating at later stages[4]. Similarly, Nodal (Spaw) may function only in specific cell populations (perhaps peripheral cells) in the zebrafish heart. In fact, *spaw* shows higher expression in the lateral (peripheral) region of the LPM (cardiac disc)[26,28]. Additionally, a Nodal antagonist, *lefty2*, is expressed in a pattern complementary to that of *spaw*: higher expression in the central region[26,28], implying limited activation of Nodal signaling in the central/inner region of the disc. More detailed analysis of *spaw* and *lefty2* expression is required to address this possibility. Analysis of cell behavior in association with reporter lines for *spaw* or its downstream

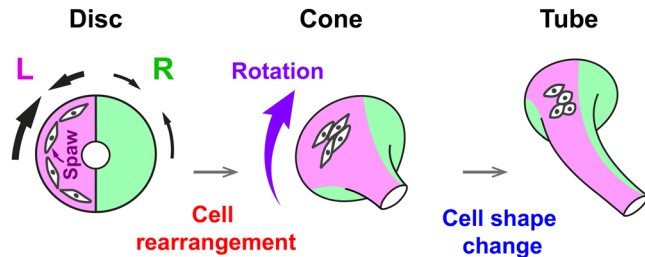

**Fig. 9 Diagram summarizing the process of heart tube formation in zebrafish.** The disc-to-tube transformation occurs through convergent extension, initially driven by oriented cell rearrangement and subsequently by cell shape change. Both cell behaviors occur circumferentially toward the anterior seam of the left (L) and right (R) primordia (black arrows), converging the disc anteriorly along the circumferential direction. Nodal (Spaw) signaling promotes rearrangement and shape changes of left cells, leading to more rapid convergence of the left primordium, and thereby rotating the disc (purple arrow).

effectors such as phosphorylated Smad2/3 would help to resolve this question. Alternatively, the results may reflect different developmental timings between peripheral and inner regions. As mentioned above, the disc-to-tube transformation/cell shortening occurs progressively from the central/inner to peripheral sides of the disc. Therefore, by analyzing earlier stages, using a genetic driver that enables earlier GFP expression in myocardium, left-right differences may also be found in the inner region. Likewise, the lesser amount of cell rearrangement observed in the inner region, as compared with that in the peripheral region, also might be attributed to different developmental timings between these regions. To partially test whether differences between peripheral and inner regions reflect differences in developmental timings, we realigned the measurements of Figs. 4e, 5c, 6d in time based on the initiation of cell shortening. The realigned results still showed significant left-right differences only in the peripheral region (Supplementary Figs. 11b, c and 12b, c), consistently with the original results (Figs. 5c and 6d). The realignment of results of Fig. 4e showed similar, but not identical, patterns between peripheral and inner cells: inner cells displayed more cell length reduction (Supplementary Figs. 11a and 12a), implying different cell properties between these regions. However, analyses of earlier stages are required to fully examine these possibilities.

The left-right differences in cellular and tissue behaviors we observed as the heart tube extended/rotated asymmetrically could either arise indirectly from asymmetric deformation of the heart tube or from intrinsic left and right cell properties. To address this, we analyzed non-fused hearts, showing that cellular and tissue dynamics in cardia bifida are essentially the same as that in normal hearts: the left and right primordia converged asymmetrically, and the more rapid convergence of the left primordium is initially driven by cell rearrangement and subsequently by cell shortening. The results demonstrate that these changes are not caused by external mechanical forces, but are controlled by intrinsic genetic programs.

Compared with normal hearts, cell shortening in both the left and right primordia of non-fused hearts tended to be delayed. This might be due to defective endoderm, which functions as a substrate for myocardial cells to change their shape. Alternatively, the loss of a counterpart of the paired heart primordia might have changed physical conditions within the primordia. During normal fusion, the paired heart primordia move and converge anteriorly toward each other and collide at the junction. Our live imaging showed that the anterior border of the left and right primordia did not move much during heart rotation, whereas the posterior border shifted clockwise, implying that forces generated by anteriorward movement of left and right cells might be

balanced at the anterior border (additionally, we speculate that the low ability of right cells for exchanging neighbors might restrict left cells to intercalate into right cells beyond the anterior border). Thus, the colliding left and right cells may function as mechanical constraint on the anterior movement of the counterpart cells of one another. In cardia bifida, in the absence of such constraint, efficient cell shortening might be compromised. Or the delay of cell shortening simply may be attributed to differences in cell populations analyzed (posterior cells are also included in the analyses of cardia bifida).

In zebrafish, the paired heart primordia generate a circumferentially-closed disc by fusion[21,23,58], whereas in the chick and mouse, fusion initiates at a single point, and progresses bidirectionally[20,30,59,60]. This study, together with our previous work[30], showed that despite the different manners of fusion, the flat heart primordia transform into a tube through CE both in the chick and fish. The directions of the convergence (anteroposterior/circumferential direction in the LPM/disc) and extension (medial-lateral direction) were consistent between these species, suggesting evolutionary conservation of this process. Therefore, we consider that cell behaviors underlying early heart morphogenesis and the effect of Nodal signaling on these behaviors documented here can be commonly applied to vertebrate heart tube morphogenesis. We showed that in zebrafish, circumferential convergence converts the disc-shaped primordium into a tube, with the stronger convergence on the left side, promoted by Nodal signaling, rotating the disc clockwise. In the chick, the first heart fusion and CE form the initial heart tube in the midline. Through subsequent CE, the unfused flat primordia incorporate into the both poles of the initial heart tube. *Nodal* is shown to be expressed in this second population in mice, generating asymmetry at the poles[4]. If the left-sided Nodal signaling promotes CE in this cell population, that would lead more rapid incorporation and elongation on the left side at the heart poles, which would orient the heart tube toward the right[30]. In concordance with this, a previous study in the chick showed that cells of right origin constitute 55.5% of all myocardial cells in the pre-looping heart tube (HH10−), but only 48% when C-looping completed (HH11-11+), suggesting more incorporation of left cells during C-looping[61], where both arterial and venous poles extend toward the right[16]. Contrary to this, in mice the venous pole undergoes a leftward displacement with more cell ingression on the right side[4,20]. This may be a species difference or a leftward displacement of the venous pole in the chick may occur after C-looping. Thus, we propose that Nodal signaling may similarly enhance CE in other vertebrates, driving asymmetric heart tube deformation and rotation, as we observed in zebrafish.

## Materials and methods

**Zebrafish**. Zebrafish (*Danio rerio*) were maintained under standard condition. All experiments using zebrafish were approved by the Animal Experimentation Committee of the National Cerebral and Cardiovascular Center and the Doshisha University. Transgenic line *Tg(myl7:EGFP-CAAX)^ncv536Tg* was established by co-injecting pTol2 vector-based plasmids[62,63], containing the *myl7* promoter[31] and the enhanced green fluorescence protein (EGFP) gene fused to the CAAX motif, with Tol2 transposase mRNA into wild-type (AB) embryos at one- to two-cell stage. Embryos were raised at 26–28.5 °C and staged according to hours post-fertilization (hpf) and morphology[64]. The transgenic line *Tg(myl7:EGFP-CAAX)^ncv536Tg* is available from the corresponding author upon request.

**Confocal time-lapse imaging and image analysis**. Embryos were dechorionated and mounted in 0.8% low-melting agarose dissolved in E3 medium containing 0.016% tricaine (Ethyl 3-aminobenzoate methanesulfonate, Sigma) in 35 mm plastic dishes. The agarose-mounted embryos were covered with 0.03% sea salt (REI-SEA marine, Iwaki) water containing 0.016% tricaine to immobilize the embryos. Images were acquired from the dorsal side of embryos using an upright Olympus FV1000 or FV1200 laser-scanning confocal microscope with water-immersion ×20 objective (XLUMPLFLN 20X W, NA:1.00) with ×1.4–2.0 zoom, 10 min time intervals, 2.0 µm z-step, 30–50 z-slices for each sample.

The confocal 4D data sets were volume rendered using FluoRender software (https://www.sci.utah.edu/software/fluorender.html). During the volume rendering with FluoRender, gamma values of fluorescent signals have been changed. Individual cells were tracked using manual tracking plugin of ImageJ. The length of cell arrays, individual cells (long axis), and the overlap between cells were measured on the apical side of cells using ImageJ (NIH). The lengths of peripheries of the left/right primordia (Supplementary Fig. 3c) and the width of the heart tube (Supplementary Fig. 3f) were similarly measured with ImageJ. These measurements were performed in 2D projection. It should be noted that in case the plane where the length is measured is obliquely oriented to the horizontal, the accuracy of the measured value can be compromised. The perpendicular lengths (Supplementary Figs. 1d and 3g) were measured in 3D with FluoRender using Two-point ruler tool because the perpendicular axis of the heart tube can be inclined with respect to the horizontal plane. During heart tube formation, the heart primordia undergo three-dimensional morphological changes. Such 3D changes cannot be entirely analyzed in 2D measurements. It should be noted that this study focuses morphological changes of limited regions of heart primordia in limited time windows. At early stages, GFP signals in the peripheral region of the cardiac disc were often too weak while GFP signals in the inner region were sufficient. Therefore, there is a 1-2 h lag in the initial time points of the measurements between peripheral and inner cell arrays. Peripheral and inner cell arrays were identified and used in our study because there is no landmark to determine absolute positions within the cardiac disc. We selected cells at the most outer position of the cardiac disc, when the disc has just formed, as A3 cell arrays, and cells at more inner positions as A2-A1 cell arrays in a sequential manner. Newly GFP-expressing cells incorporated into the peripheries of A3 cell arrays in the next 1–2 h, which were selected as A4 cell arrays. To classify left and right cells, we utilized cell tracking to identify their left/right origins. In most cases, we began time-lapse recording before the left and right heart primordia fused, so we could easily identify left-right origins. In some cases, the left and right heart primordia partly fused (fusion starts from the posterior/central to anterior/peripheral sides) when the recording began, and identification of left-right origins of cells in the posterior regions were sometimes difficult. In such cases, we did not determine the origin and showed them as "cells with ambiguous left-right origins" as shown in Fig. 7b (colored cells with white). For the left-right comparison, cells of the same number, circumferentially arrayed in a row in similar relative positions between the left and right heart primordia, were carefully chosen for measurement. At least three embryos were analyzed for each comparison.

**Morpholino injection.** The previously validated morpholino oligonucleotides (Gene Tools), 4–7 ng of *s1pr2* MO[38,40,43] and 5 ng of *spaw* MO[15,24,28] were injected into one to two cell stage embryos. The sequence of the MOs: *spaw* MO: 5'-TGGTAGAGCTTCAACAGACTCTGCA-3', *s1pr2* MO: 5'-CCGCAAACAGAC GGCAAGTAGTCAT-3'.

**Whole-mount in situ hybridization.** Embryos were fixed with 4% paraformaldehyde (PFA) overnight at 4 °C, rinsed with phosphate-buffered saline containing 0.1% Triton X-100 (PBTx), and dehydrated through a graded series of methanol/PBTx on ice. Embryos were gradually rehydrated into PBTx, incubated in 6% $H_2O_2$ for 1 h at room temperature (RT), followed by incubation in 10 μg/ml proteinase K for 10 min at RT, and post-fixed in 4% PFA containing 0.2% glutaraldehyde for 20 min at RT. Embryos were pre-incubated in hybridization buffer at 65 °C for 1 h and then incubated with DIG-labeled RNA antisense probes at 65 °C overnight. After washing, embryos were incubated with alkaline phosphatase conjugated anti-digoxigenin antibody (Roche, 1:2000 dilution) at 4 °C overnight. After extensive washing, staining reactions were carried out with BM purple (Roche) to visualize the signals. Embryos were cleared through a graded series of glycerol/PBTx and imaged using a stereo microscope.

**Statistics and reproducibility.** Data are presented as mean ± s.d. (standard deviation). To compare convergence rates between the left and right primordia, a two-tailed *t*-test assuming unequal variances was used (Excel). Statistical analysis shown in Supplementary Fig. 3c, f, g was performed using GraphPad Prism 9 with an unpaired two-tailed *t*-test with Welch's correction. At least three independent experiments were performed for each analysis. *p* values < 0.05 were considered statistically significant.

**Reporting summary.** Further information on research design is available in the Nature Research Reporting Summary linked to this article.

## Data availability
All data supporting the findings of this study are included in this article. The source data underlying the graphs presented in the main figures of this study are provided in Supplementary Data 1.

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

## Acknowledgements

We thank Naoki Mochizuki (National Cerebral and Cardiovascular Center Research Institute, Japan) for valuable and thoughtful discussions and generous supports. We also thank Sayuri Yonei-Tamura (Tohoku University, Japan) and Brent Bisgrove (University of Utah, USA) for critical reading of the manuscript and helpful discussions. Research reported here was supported by the Japan Society for the Promotion of Science (JSPS) Grants-in-Aid for Scientific Research (KAKENHI) Grants (JP22K06820 to H.K.), and the Promotion and Mutual Aid Corporation for Private Schools of Japan (PMAC) Scholarship Fund for Women Researchers (to H.K.).

## Author contributions

H.K., Y.S. and G.C.S. conceived the study, H.K. designed and performed the experiments and analyzed the results. H.K. wrote the original draft of the paper. H.K., G.C.S. and Y.S. edited the manuscript.

## Competing interests

The authors declare no competing interests.
