## [Peer Review File · Communications Biology]

Reviewers' comments:

Reviewer #1 (Remarks to the Author):

In this article, authors address the cellular mechanisms of heart tube formation, using the zebrafish as a model amenable to live-imaging. Based on careful cell segmentation, they quantify cell rearrangements and cell shape changes during the transition from the disc to the tube. They identify a spatio-temporal dynamic of left-right asymmetries in these processes. They elegantly exploit a model of cardia bifida, to assess left-right asymmetries independently of tubulogenesis. Knock-down of Nodal/spw in this condition suggests that left-right asymmetries depend on Nodal signalling.

Anomalies in left-right asymmetry of the embryonic heart tube is at the origin of severe congenital heart defects, with misalignment of cardiac chambers. Yet, the cellular mechanisms underlying this process remain poorly understood. The approaches used here with high resolution live-imaging are well thought, providing overall significant advance into asymmetric heart tube formation. I have a few comments to clarify some of the claims.

Major comments

-Quantification of cell behavior is performed in one direction, showing circumferential convergence. However, the claim of extension is not supported by quantifications, neither for cell rearrangements nor for cell shape changes. Quantification at the level of the tissue should be added : what is the relative circumference and perpendicular length of the left region at the beginning and end of the movie ? The surface proportion is indicated as "roughly" but should be quantified in different embryos.

-The claim that "convergence of the cardiac disc is primarily driven by oriented cell rearrangement and subsequently by cell shape change" is overstated. This would require to quantify the contribution of each cell behaviour to the tissue shape change, or test functionally the impact of removing specific cell behavior.

-The claim that "the role of Nodal is not to drive CE itself, but to enhance the intensity of CE" is not supported by experiments. Line 342, authors refer to Fig 5C and 6C which do not manipulate Nodal signaling. Fig. 7 instead shows that convergence is "abolished" (line 293) in *s1pr2* ; *spaw* double morphants, if not reversed ("increase in length"). Analysis of single *spaw* morphants is required, to conclude whether Nodal initiates or amplifies asymmetries in cell behaviour. This conclusion also requires statistical test of whether measures are significantly different from the zero line. How comparable are the measured values acquired in different experimental conditions, given the normalization ? Cell segmentation in association with a Nodal reporter line would permit to conclude whether differences between peripheral and central cell arrays reflect transient Nodal signaling.

Minor comments

-Please clarify the contradiction between comments of figure 1 "myocardial cells in the cardiac disc undergo cell-cell intercalation" and the discussion line 351 "changes in positional relationship between neighboring cells were modest in our time-lapse imaging."

-To test whether peripheral and central cells reflect different timings, measures could be realigned in time based on the initiation of cell shortening.

-the species analysed should be included in the title

-Yue et al 2004 does not report processes downstream of Nodal

-line 429 "probably in the mouse as well" : authors can refer to PMID: 29202929 or cited ref Le Garrec 2017

-line 447-451 contradict the finding that left-right asymmetries are opposed at the arterial and venous poles (Le Garrec, 2017 ; Desgrange et al 2020)

-line 313, 343 : reference to Desgrange 2020 is required, as the first demonstration that Nodal has an amplifying role

- Methods : please add the total image volume and explain if segmentations are done on a 2D projection or a 3D image. If relevant, the limitation of analyzing complex shape changes in a 2D projection should be acknowledged. Please clarify also that cell shape/shortening has been measured only on the apical side.
- Methods : please define which criteria are used to classify cells as peripheral and central, to classify cells as left/right, to select cell arrays.
- Statistics : the number of observations should be specified in all figure legends, with the unit (in cells ? arrays ? embryos ?). Units are also lacking in the text.
- why are cells restricted in their rotation at the anterior ? please comment in the discussion

Reviewer #2 (Remarks to the Author):

In the manuscript by Hinako Kidokoro and colleagues, the authors investigate how left-asymmetric Nodal signaling impacts cell behavior in the cardiac disc that will give rise to the asymmetrically folded heart tube during heart formation in zebrafish. Although it is known that the Nodal cascade plays a role in asymmetric heart formation, and asymmetric heart jogging of the heart tube, it is yet unclear how cells in the heart tube-preceding tissue – the cardiac disc – asymmetrically contribute to the forming tube. By advanced time lapse imaging employing myocard specific membrane labeling and single cell tracking and shape description, the authors find a circumferential convergence with higher mobility and flexibility on the left side under the control of the Nodal cascade. In an elegant experiment in which the authors block fusion of the lateral plate mesoderms (LPM) in the midline, the left side (under the control of Nodal) shows the same higher activity/mobility/shapechanging of cells compared to the right LPM. Loss of Nodal signaling inhibits this motion and shape changing.

This study is carefully undertaken and experiments follow a logic order. Novelty is given as the authors provide detailed analysis at the single cell level with respect of position within the developing cardiac disc and contribution to the heart tube. This study integrates and cites relevant work from the field where appropriate. Implications with planar cell polarity and biophysical mechanistic renders it interesting also for a broader field of biologists.

The experiments provided all address the claims raised. The manuscript is very well written, easy to understand and the figures provided are of high quality and are illustrated to understand the morphodynamic processes described in the main text. The same good quality is found in all supplementary movies. I hence have no objections and fully support publication of this manuscript.

Minor points:

Line 78: ... by the rotation – at this point it is unclear which rotation the authors refer to, I suggest "...by a rotation ..." instead, or even without "a".

Reviewer #3 (Remarks to the Author):

In this study, Kidokoro et al. make use of a Tg(myf7:EGFP-CAAX) zebrafish reporter line to present a detailed 2D analysis of the cardiomyocyte behavior during cardiac jogging in the zebrafish. They describe which cardiomyocyte rearrangements and shape changes accompany the transformation of the cardiac disc into the leftward-extending linear tube during the cardiac jogging process. In the last part of the study the authors use morphants to study these same cellular behaviors in non-fusing hearts with or without Nodal-mediated Left-Right (LR) information.

The study is interesting as it sheds more light on how tissue morphogenesis in the embryo is influenced by left-right patterning. The positioning of the heart tube at the left side is the first morphological sign of LR asymmetry in the embryo and can be studied very well in zebrafish due to its optical transparency. The study bases itself on high quality imaging. The effort to describe the cellular rearrangements and shape changes in the jogging heart is commendable and the

proposed role of convergence-extension movements is interesting and novel. However, the study remains largely descriptive and is limited to analysis in 2D rather than in 3D.

These are my major points:

-The introduction of the manuscript is rather long and focuses largely on cardiac looping, which in zebrafish is considered a distinct morphogenetic event from cardiac jogging, on which the authors experimentally focus. The authors cite many of the studies belonging to the relevant literature on cardiac jogging in zebrafish, so this could be exploited in a more "jogging-specific" introduction, leaving out parts of the lengthy initial paragraphs on cardiac looping.

-A clear limitation of the study is that the analysis is conducted in 2D, while jogging is actually a process in which the heart undergoes a major 3-dimensional transformation. In the experimental set-up, it is not possible to analyze left- and right-originating cardiomyocytes equally as the latter become "hidden" quite quickly during jogging. The authors acknowledge this issue and limit their analysis to 'visible' cells. However, this 2D analysis becomes a clear limitation when analyzing the contribution of the left- and right-derived cells related to figure 4. The conclusion that left-originating cells constitute only about 1/3 of the heart tube (Figure 5) is surprising and in disagreement with previously studies cited by the authors, in which it is demonstrated that the LR origin of the cardiomyocytes in the cardiac disc corresponds quite accurately to the upper and lower "halves" of the heart tube at the end of jogging. In addition, for such an important reduction in surface area, one would expect that left-derived cells are lost (NB this is refuted by the authors at lines 156-158, albeit without quantification) or become much smaller during heart tube formation, and none of these are shown here. It is not very clear how the authors come to this conclusion and how it was analysed.

-In the last part of this study the authors address whether the asymmetric CE movements that they observed are the consequence of left-sided Nodal signaling. They perform these experiments in the context of non-fused heart fields. How do the heart tubes in embryos without cardiac fusion look like? Are there differences in shape between the left and right tube within the same embryo? It would be very important to know whether the asymmetric CE movements described in Figures 3 and 4 are also lost in a context in which the heart tube is formed (spaw mutant background or spaw KD) but remains at the midline (jogging is disrupted) and what is the consequence for the shape of the heart tube (in terms of length and width of the tube). Related to this, have the authors tried to correlate the asymmetric CE movements to the direction of jogging (for example in reverse joggers due to right-sided Nodal activation? This can be achieved either by implanting Nodal beads in spaw KD or using cilia mutants with randomized jogging).

Minor Points:

-Relatively to the entire cardiac disc, the definition of peripheral and central cardiomyocytes (Figures 4 and 5) could be improved or better defined. The central cardiomyocytes in particular seem actually rather peripheral.

-The discussion is generally lengthy and too speculative. Especially the paragraph "regional difference within the cardiac disc" is based on observations that are not subtended by regional markers of the cardiac disc.

-Line 292: the cited literature is not on zebrafish.

-Line 490: typo

We sincerely thank the three reviewers for their careful reading of the manuscript and their thoughtful and constructive comments. We feel that the manuscript has been significantly strengthened by addressing each of all reviewers' queries, as detailed in a point-by-point response below. Especially, adding the new data on the Spaw single-knockdown has clarified the role of Nodal signaling in the asymmetric morphogenesis analyzed in this study. We hope that the reviewers agree. We again are grateful for having this opportunity to incorporate the views of the expert reviewers into our study.

Reviewers' comments:

Reviewer #1 (Remarks to the Author):

In this article, authors address the cellular mechanisms of heart tube formation, using the zebrafish as a model amenable to live-imaging. Based on careful cell segmentation, they quantify cell rearrangements and cell shape changes during the transition from the disc to the tube. They identify a spatio-temporal dynamic of left-right asymmetries in these processes. They elegantly exploit a model of cardia bifida, to assess left-right asymmetries independently of tubulogenesis. Knock-down of Nodal/spw in this condition suggests that left-right asymmetries depend on Nodal signalling.

Anomalies in left-right asymmetry of the embryonic heart tube is at the origin of severe congenital heart defects, with misalignment of cardiac chambers. Yet, the cellular mechanisms underlying this process remain poorly understood. The approaches used here with high resolution live-imaging are well thought, providing overall significant advance into asymmetric heart tube formation. I have a few comments to clarify some of the claims.

Major comments

1. -Quantification of cell behavior is performed in one direction, showing circumferential convergence. However, the claim of extension is not supported by quantifications, neither for cell rearrangements nor for cell shape changes. Quantification at the level of the tissue should be added : what is the relative circumference and perpendicular length of the left region at the beginning and end of the movie ?

[RESPONSE]

We have added data on the relative circumference and perpendicular length of the left region at the beginning and end of the movie as supplementary Figure 1. These new data show that the tissue extends by 80 % in the perpendicular direction while converging by 50 % circumferentially (line 118-120).

2. The surface proportion is indicated as “roughly” but should be quantified in different embryos.

[RESPONSE]

We presume that the reviewer refers to the description for Figure 5. First, we apologize for our imprecise description. As the reviewer suggested, we should not have stated the proportion of the left primordium to the entire cardiac disc without quantification. Because our focus here is to identify left-right differences that drive rotation of the cardiac disc, we have chosen to quantify the circumferential length of each left and right primordium rather than the surface area for the following reasons:

1. While the heart primordia converge circumferentially, they also undergo extension in the perpendicular direction. Considering the nature of convergent extension, the surface area may not change significantly, even when one axis greatly narrows or extends. For a potential driver of the rotation of the cardiac disc, we consider that the left-right asymmetry in the circumferential convergence would be more important than asymmetry in the surface area.

2. Accurate quantification of the surface area is technically very difficult because involuted parts of the right primordium are hidden behind the tubular heart and the obtained images at these regions are blurry. In addition, the involuted parts of the right primordium are usually in close contact, which makes it difficult to quantify the surface area.

In the revised manuscript, we corrected the description as follows (line 192-200).

-Our cell tracking showed that at this stage left and right heart primordia equally compose each half of the cardiac disc (Fig. 5a, 1 h). At this point, the anterior and posterior borders of the left and right primordia were at 12 and 6 o'clock, respectively. However, as the heart tube formed, the posterior border shifted to 7 o'clock (Fig. 5a, 5 h), and then to 9 o'clock (Fig. 5a, 9.5 h, n=8/8 embryos), suggesting that the left primordium converged more rapidly than the right one. We quantified the circumferential length of the peripheries of the left and right primordia (Fig. S2d, e): the ratio of the left peripheral length to the right one was

initially 1: 1 ± 0.04 (0 h, Fig. S2e), subsequently became 1: 0.67 ± 0.04 at 4 hours, and 1: 0.5 ± 0.04 at 8.5 hours (n=3 embryos), supporting our notion.

We confirmed that the clockwise shift of the posterior border of the left and right primordia toward 7-9 o'clock direction is consistently observed in different embryos (n=8/8 embryos).

We also would like to remind the reviewer that we quantified the convergence of the anterior portion of the primordium (Fig. 5c), showing that the left primordium in the anterior region converged more rapidly than that of the right primordium.

3. -The claim that “convergence of the cardiac disc is primarily driven by oriented cell rearrangement and subsequently by cell shape change” is overstated. This would require to quantify the contribution of each cell behaviour to the tissue shape change, or test functionally the impact of removing specific cell behavior.

[RESPONSE]

We agree with the reviewer's comment. We have reworded the sentence to “Convergence of the cardiac disc is **initially** driven by oriented cell rearrangement and subsequently by cell shape change” in the revised manuscript (line 147-148).

4. -The claim that “the role of Nodal is not to drive CE itself, but to enhance the intensity of CE” is not supported by experiments. Line 342, authors refer to Fig 5C and 6C which do not manipulate Nodal signaling. Fig. 7 instead shows that convergence is “abolished” (line 293) in *s1pr2* ; *spaw* double morphants, if not reversed (“increase in length”). Analysis of single *spaw* morphants is required, to conclude whether Nodal initiates or amplifies asymmetries in cell behaviour. This conclusion also requires statistical test of whether measures are significantly different from the zero line. How comparable are the measured values acquired in different experimental conditions, given the normalization?

[RESPONSE]

We appreciate the reviewer's thoughtful and constructive comment. We have added the analysis of single *spaw* morphants in Fig. 7, Figs. S2 and S3 (line 263-283). In *Spaw* knockdown embryos, the more rapid convergence of the left peripheral region than the right corresponding region, through more active cell rearrangement/cell shortening was abolished, leading to loss of left-right asymmetry in the cardiac convergence, as well as the heart

rotation. Comparison between uninjected and *spaw* MO-injected embryos revealed that *Spaw* knockdown significantly reduced the convergence of the left primordium in its peripheral region, whereas it had no significant effect on the inner (central) region nor on the entire right primordium. These results suggest that *Spaw* increases the extent of convergence in the peripheral region of the cardiac disc, consistently with the results from left-right comparison in normal (Fig. 5) and non-fused (Fig. 6) hearts.

We also compared the length and width of the heart tube between normal and *spaw*-knockdown embryos in response to the concern of reviewer 3. The results showed that the venous end (peripheral side) of the heart tube was widened in the *spaw*-morphants, whereas the widths of the arterial half of the tube were not significantly different from those of normal embryos (Fig. S2h) The length of the heart tube is significantly reduced in *spaw*-morphants (Fig. S2i). These results are concordant with the results described above (Fig. 7 and Fig. S3).

Although we carefully select cells to analyze in similar relative positions in nearly equivalent developmental stages between embryos, they cannot be identical because there are variabilities in cell number and morphological dynamics of heart primordia from embryo to embryo. Hence, we consider that comparison between left and right within the same embryo would be more reliable than the comparison between different embryos (normal vs *Spaw* knockdown). We therefore show the left-right comparison in single *Spaw* knockdown embryos as a main figure (Fig. 7), and comparison between normal and *Spaw* knockdown embryos as a supplementary figure (Fig. S3) in the revised manuscript. It should be noted that both comparisons show consistent results.

5. Cell segmentation in association with a Nodal reporter line would permit to conclude whether differences between peripheral and central cell arrays reflect transient Nodal signaling.

[RESPONSE]

We thank the reviewer for this suggestion. Considering Nodal is a secreted protein, a reporter line for downstream effector of Nodal (such as p-smad) may be required. As we currently do not have such reporter line nor a Nodal reporter line, we have added these important ideas to the Discussion so that they can be considered in a future study (line 372-374).

-Analysis of cell behavior in association with reporter lines for *spaw* or its downstream effectors such as phosphorylated Smad2/3 would help to resolve this question.

Minor comments

1. -Please clarify the contradiction between comments of figure 1 “myocardial cells in the cardiac disc undergo cell-cell intercalation” and the discussion line 351 “changes in positional relationship between neighboring cells were modest in our time-lapse imaging.”

[RESPONSE]

Thank you for the suggestion. Line 351 refers to right cells whose cell-cell intercalation is less active as compared to left cells. We have clarified this in line 339-341 of the Discussion.

2. -To test whether peripheral and central cells reflect different timings, measures could be realigned in time based on the initiation of cell shortening.

[RESPONSE]

We appreciate the reviewer’s suggestion. We performed realignment of the measurements and added the results as follows in the Discussion (line 381-388).

- To partially test whether differences between peripheral and inner regions reflect differences in developmental timings, we realigned the measurements of Fig. 4c/5c/6c based on the initiation of cell shortening. The realigned results still showed significant differences only in the peripheral region (Fig. S4b-c), consistently with the original results (Fig.5c, 6c). The realignment of results of Fig. 4c showed similar, but not identical, patterns between peripheral and inner cells: inner cells displayed more cell length reduction, implying different cell properties between these regions. However, analyses of earlier stages are required to fully examine these possibilities.

3. -the species analysed should be included in the title

[RESPONSE]

We thank the reviewer for the suggestion. We have considered it, but decided to not include the species analyzed in the title, and instead to state it in the Abstract (line 31). We do not consider that the findings of our study are specific to zebrafish as our data, together with our previous work in the chick, suggest that the basic cellular mechanisms underlying early heart morphogenesis are well conserved across species. Therefore, we hope to avoid a title that may lead to misleading impression that the scope of the study is to understand fish specific

heart morphogenesis.

4. -Yue et al 2004 does not report processes downstream of Nodal

[RESPONSE]

Thank you for pointing this out. We have deleted the citation from the sentence (line 64).

5. -line 429 “probably in the mouse as well” : authors can refer to PMID: 29202929 or cited ref Le Garrec 2017

[RESPONSE]

We thank the reviewer for the suggestion. We have cited both papers in the corresponding part (line 416-417).

6. -line 447-451 contradict the finding that left-right asymmetries are opposed at the arterial and venous poles (Le Garrec, 2017 ; Desgrange et al 2020)

[RESPONSE]

During C-looping in the chick, both the arterial and venous poles extend toward the right, making a “C”-shape (please note that the junction of the left and right primordia at the venous pole moves to the right side of the midline (notochord) during C-looping in Fig.1 and Video S2 in Kidokoro et al., 2008). This rightward extension of the venous pole also occurs when the arterial half of the heart tube was ablated (Fig. 6, Video S4 in Kidokoro et al., 2008). We also confirmed using 3D imaging that the dorsal margins of the left right primordia (yellow arrows in the image below) are both located at slightly on the right side (n=5 embryos, unpublished data, The red line in the left image indicates the position where the transverse section was made). These data suggest that in the chick, the leftward displacement of the venous pole does not occur during C-looping. We speculate that it may occur later stages after C-looping (please see the picture of Fig. 1H in Manner 2000), to generate a helical loop as reported in the mouse, although we have not examined these later stages. We consider that this difference between the chick and mouse may be a timing difference of the venous pole displacement, and it may relate to the differences of the distance between the arterial and venous poles between species (longer in the chick (Le Garrec, 2017)). We have added discussion about the contradiction that the reviewer pointed out in the Discussion (line 435-437).

[data redacted]

Reference:

Männer, 2000. *Anat. Rec.* 259, 248–262

Kidokoro, et al., 2008. *Dev. Dyn.* 237, 3545–3556.

7. -line 313, 343 : reference to Desgrange 2020 is required, as the first demonstration that Nodal has an amplifying role

[RESPONSE]

We have added the reference to line 307 (originally line 313) as the reviewer suggested. The description in line 343 of initially submitted manuscript has been changed along with the data addition of *spaw* single knockdown analysis. The corresponding part of the revised manuscript is as follows (line 333-335):

- These results suggest that the peripheral cells have basal levels of cell rearrangement and cell shape change, and that the role of Nodal is to locally enhance these levels.

8. -Methods : please add the total image volume and explain if segmentations are done on a 2D projection or a 3D image. If relevant, the limitation of analyzing complex shape changes in a 2D projection should be acknowledged. Please clarify also that cell shape/shortening has been measured only on the apical side.

[RESPONSE]

We have added this information and limitations in the Methods/Discussion of the revised manuscript as follows. While we measured the cell length/overlap length/the length of cell arrays in 2D in the analysis of circumferential convergence, the perpendicular length of the tissue/heart tube was measured in 3D as the perpendicular axis can be inclined with respect to the horizontal plane, which may affect the accuracy of the measured length. We also added this explanations to the Methods.

-Images were acquired from the dorsal side of embryos using an upright Olympus FV1000 or FV1200 laser-scanning confocal microscope with water-immersion 20X objective (XLUMPLFLN 20X W, NA:1.00) with 1.4-2.0X zoom, 10 min time intervals, 2.0 mm z-step, 30-50 z-slices for each sample. (line 458-462)

- The length of cell arrays, individual cells (long axis), and the overlap between cells were measured on the apical side of cells using ImageJ (NIH). The lengths of peripheries of the left/right primordia (Fig. S2e) and the width of the heart tube (Fig. S2h) were similarly measured with ImageJ. These measurements were performed in 2D projection. It should be noted that in case the plane where the length is measured is obliquely oriented to the horizontal, the accuracy of the measured value can be compromised. The perpendicular lengths (Figs. S1d and S2i) were measured in 3D with FluoRender using Two-point ruler tool because the perpendicular axis of the heart tube can be inclined with respect to the horizontal plane. During heart tube formation, the heart primordia undergo three-dimensional morphological changes. Such 3D changes cannot be entirely analyzed in 2D measurements. It should be noted that this study focuses morphological changes of limited regions of heart primordia in limited time windows. (Line 466-477)

9. - Methods : please define which criteria are used to classify cells as peripheral and central, to classify cells as left/right, to select cell arrays.

[RESPONSE]

We have added this information to the Methods as follows (line 480-492).

-Peripheral and inner cell arrays were identified and used in our study because there is no landmark to determine absolute positions within the cardiac disc. We selected cells at the most outer position of the cardiac disc, when the disc has just formed, as A3 cell arrays, and cells at more inner positions as A2-A1 cell arrays in a sequential manner. Newly GFP-

expressing cells incorporated into the peripheries of A3 cell arrays in the next 1-2 hours, which were selected as A4 cell arrays. To classify left and right cells, we utilized cell tracking to identify their left/right origins. In most cases, we began time-lapse recording before the left and right heart primordia fused, so we could easily identify left-right origins. In some cases, the left and right heart primordia partly fused (fusion starts from the posterior/central to anterior/peripheral sides) when the recording began, and identification of left-right origins of cells in the posterior regions were sometimes difficult. In such cases, we did not determine the origin and showed them as “cells with ambiguous left-right origins” as shown in Fig. 7a’ (colored cells with white).

10. - Statistics : the number of observations should be specified in all figure legends, with the unit (in cells ? arrays ? embryos ?). Units are also lacking in the text.

[RESPONSE]

We have added the number of observations with the unit in all figure legends and the main text.

11. -why are cells restricted in their rotation at the anterior? please comment in the discussion

[RESPONSE]

We appreciate the reviewer’s suggestion. We consider that this is an important point. For left cells to move beyond the anterior midline of the disk toward the right side, either left cells are required to intercalate between right cells, or right cells are required to move posteriorly. We speculate that intercalation of left cells into right cells is restricted due to a lower ability of right cells to exchange neighbors. As for the latter, we observed that during heart fusion, both left and right cells in the anterior region moved towards the anterior midline, consequently, the left and right cells collided at the midline. Considering that the border of the left and right primordia did not shift to either the left or right much in our observation, forces generated by the anterior movements of the left and right cells at the border might be balanced. We have added these notions to the Discussion as follows (line 402-408).

- During normal fusion, the paired heart primordia move and converge anteriorly towards each other and collide at the junction. Our live imaging showed that the anterior border of left and right primordia did not move much during heart rotation, whereas the posterior

border shifted clockwise, implying that forces generated by anteriorward movement of left and right cells might be balanced (in addition, we speculate that the low ability of right cells for exchanging neighbors might restrict left cells to intercalate into right cells beyond the anterior border).

Reviewer #2 (Remarks to the Author):

In the manuscript by Hinako Kidokoro and colleagues, the authors investigate how left-asymmetric Nodal signaling impacts cell behavior in the cardiac disc that will give rise to the asymmetrically folded heart tube during heart formation in zebrafish. Although it is known that the Nodal cascade plays a role in asymmetric heart formation, and asymmetric heart jogging of the heart tube, it is yet unclear how cells in the heart tube-preceding tissue – the cardiac disc – asymmetrically contribute to the forming tube. By advanced time lapse imaging employing myocard specific membrane labeling and single cell tracking and shape description, the authors find a circumferential convergence with higher mobility and flexibility on the left side under the control of the Nodal cascade. In an elegant experiment in which the authors block fusion of the lateral plate mesoderms (LPM) in the midline, the left side (under the control of Nodal) shows the same higher activity/mobility/shapechanging of cells compared to the right LPM. Loss of Nodal signaling inhibits this motion and shape changing.

This study is carefully undertaken and experiments follow a logic order. Novelty is given as the authors provide detailed analysis at the single cell level with respect of position within the developing cardiac disc and contribution to the heart tube. This study integrates and cites relevant work from the field where appropriate. Implications with planar cell polarity and biophysical mechanistics renders it interesting also for a broader field of biologists.

The experiments provided all address the claims raised. The manuscript is very well written, easy to understand and the figures provided are of high quality and are illustrated to understand the morphodynamic processes described in the main text. The same good quality is found in all supplementary movies. I hence have no objections and fully support publication of this manuscript.

Minor points:

1. Line 78: ... by the rotation ... – at this point it is unclear which rotation the authors refer

to, I suggest “...by a rotation ...” instead, or even without “a”.

[RESPONSE]

We are grateful to the reviewer for deeply understanding our study and summarizing the manuscript very well. We also appreciate the reviewer’s kind comments. We have removed “the” from the original sentence (line 68-69):

-In the chick, the linear heart tube turns its original ventral midline (the boundary of the left and right primordia) to the right by rotation.

Reviewer #3 (Remarks to the Author):

In this study, Kidokoro et al. make use of a Tg(myI7:EGFP-CAAX) zebrafish reporter line to present a detailed 2D analysis of the cardiomyocyte behavior during cardiac jogging in the zebrafish. They describe which cardiomyocyte rearrangements and shape changes accompany the transformation of the cardiac disc into the leftward-extending linear tube during the cardiac jogging process. In the last part of the study the authors use morphants to study these same cellular behaviors in non-fusing hearts with or without Nodal-mediated Left-Right (LR) information.

The study is interesting as it sheds more light on how tissue morphogenesis in the embryo is influenced by left-right patterning. The positioning of the heart tube at the left side is the first morphological sign of LR asymmetry in the embryo and can be studied very well in zebrafish due to its optical transparency. The study bases itself on high quality imaging. The effort to describe the cellular rearrangements and shape changes in the jogging heart is commendable and the proposed role of convergence-extension movements is interesting and novel. However, the study remains largely descriptive and is limited to analysis in 2D rather than in 3D.

These are my major points:

1. -The introduction of the manuscript is rather long and focuses largely on cardiac looping, which in zebrafish is considered a distinct morphogenetic event from cardiac jogging, on which the authors experimentally focus. The authors cite many of the studies belonging to the relevant literature on cardiac jogging in zebrafish, so this could be exploited in a more

“jogging-specific” introduction, leaving out parts of the lengthy initial paragraphs on cardiac looping.

[RESPONSE]

We thank the reviewer for the suggestion. We have removed several descriptions for looping from the initial paragraph and have focused on cardiac jogging in the Introduction as suggested.

2. -A clear limitation of the study is that the analysis is conducted in 2D, while jogging is actually a process in which the heart undergoes a major 3-dimensional transformation. In the experimental set-up, it is not possible to analyze left- and right-originating cardiomyocytes equally as the latter become “hidden” quite quickly during jogging. The authors acknowledge this issue and limit their analysis to ‘visible’ cells. However, this 2D analysis becomes a clear limitation when analyzing the contribution of the left- and right-derived cells related to figure 4. The conclusion that left-originating cells constitute only about 1/3 of the heart tube (Figure 5) is surprising and in disagreement with previously studies cited by the authors, in which it is demonstrated that the LR origin of the cardiomyocytes in the cardiac disc corresponds quite accurately to the upper and lower “halves” of the heart tube at the end of jogging. In addition, for such an important reduction in surface area, one would expect that left-derived cells are lost (NB this is refuted by the authors at lines 156-158, albeit without quantification) or become much smaller during heart tube formation, and none of these are shown here. It is not very clear how the authors come to this conclusion and how it was analysed.

[RESPONSE]

We apologize for our vague and imprecise description. We did not quantify the surface proportion of the left and right primordia, so we have corrected the description as follows (line 192-200).

- Our cell tracking showed that at this stage left and right heart primordia equally compose each half of the cardiac disc (Fig. 5a, 1 h). At this point, the anterior and posterior borders of the left and right primordia were at 12 and 6 o'clock, respectively. However, as the heart tube formed, the posterior border shifted to 7 o'clock (Fig. 5a, 5 h), and then to 9 o'clock (Fig. 5a, 9.5 h, n=8/8 embryos), suggesting that the left primordium converged more rapidly than the right one. We quantified the circumferential length of the peripheries of the left and right primordia (Fig. S2d, e): the ratio of the left peripheral length to the right one was

initially 1: 1 ± 0.04 (0 h, Fig. S2e), subsequently became 1: 0.67 ± 0.04 at 4 hours, and 1: 0.5 ± 0.04 at 8.5 hours (n=3 embryos), supporting our notion.

We agree with the reviewer. Although our observations above suggest that the left primordium undergoes more rapid circumferential convergence, this does not simply mean that the surface proportion of the left primordium to the entire heart tube also becomes significantly smaller. Considering the nature of convergent extension, it would be more feasible that the surface area of the primordium does not change significantly even when one axis of the tissue dramatically shortens (because the tissue should extend along a perpendicular axis). In addition, our quantification showed clear left-right differences in tissue convergence in the peripheral region, but not in the inner (central) region of the disc. This suggests that the asymmetric convergence occurs either in the specific region or only in the early phase of the heart tube formation. We have clarified in the revised manuscript that we quantified the convergence of the flattened disc, but not the surface area. We also have acknowledged in the Methods (line 474-477) the limitations of analyzing three-dimensional morphological changes in 2D.

- During heart tube formation, the heart primordia undergo three-dimensional morphological changes. Such 3D changes cannot be entirely analyzed in 2D measurements. It should be noted that this study focuses morphological changes of limited regions of heart primordia in limited time windows (line 474-477).

-In the last part of this study the authors address whether the asymmetric CE movements that they observed are the consequence of left-sided Nodal signaling. They perform these experiments in the context of non-fused heart fields. How do the heart tubes in embryos without cardiac fusion look like? Are there differences in shape between the left and right tube within the same embryo? It would be very important to know whether the asymmetric CE movements described in Figures 3 and 4 are also lost in a context in which the heart tube is formed (spaw mutant background or spaw KD) but remains at the midline (jogging is disrupted) and what is the consequence for the shape of the heart tube (in terms of length and width of the tube). Related to this, have the authors tried to correlate the asymmetric CE movements to the direction of jogging (for example in reverse joggers due to right-sided Nodal activation? This can be achieved either by implanting Nodal beads in spaw KD or using cilia mutants with randomized jogging).

[RESPONSE]

We agree with the reviewer's concern. The left primordium in non-fused hearts tend to become smaller than the right one as a consequence of the more rapid circumferential convergence, similar to the normal (fused) heart, as shown in Fig. 6a' (4 h). On the other hand, the non-fused hearts also exhibit unusual characteristics, including a delay in cell shortening, differing from the normal (fused) heart, as mentioned in the initially submitted manuscript. Therefore, as the reviewer suggested, investigation of the effects of Spaw knockdown in the context of normal heart tube formation would be important.

We thus have analyzed the CE movements in single spaw MO knockdown embryos: The more rapid convergence of the left primordium in the peripheral region was abolished by Spaw knockdown (Fig. 7). This result suggests that Spaw has a striking effect on enhancing CE in the peripheral region of the heart primordium, consistently with the results from the left-right comparison in normal hearts (Fig. 5) and non-fused hearts (Fig. 6). We also analyzed the resulting shape of the heart tube as suggested: As a consequence of the reduced convergence, the venous pole (peripheral region) of the heart tube in *spaw* morphants became significantly wider as compared with that of uninjected embryos, whereas no significant difference was found in the width of the arterial half of the heart tube (Fig. S2h). The length of the heart tube in the single *spaw* morphants was significantly shorter than that of the normal heart tube (Fig. S2i). These results are described in line 263-283.

Regarding the correlation between the jogging direction and CE movements, we speculate that dominance of CE movements may determine the direction of jogging because in *spaw*-morphants, both the dominant CE of the left primordium and the leftward jogging were lost. However, we have not examined this in the experimentally-induced reversed jogging. The manipulation of CE movements also would be required to conclude whether the asymmetric CE causes directional heart jogging, and this should be addressed in future studies, perhaps when the downstream targets of Nodal signaling to regulate CE are elucidated.

Minor Points:

1. -Relatively to the entire cardiac disc, the definition of peripheral and central cardiomyocytes (Figures 4 and 5) could be improved or better defined. The central cardiomyocytes in particular seem actually rather peripheral.

[RESPONSE]

We appreciate and agree with the reviewer's suggestion. We reworded "central" to "inner" in the revised manuscript.

2. -The discussion is generally lengthy and too speculative. Especially the paragraph "regional difference within the cardiac disc" is based on observations that are not subtended by regional markers of the cardiac disc.

[RESPONSE]

We appreciate the reviewer's suggestion. We have improved the Discussion to be more concise. We agree with the reviewer's concern. More detailed analysis is required to reveal the regional activity of Nodal signaling. We have added the following text in the Discussion (line 371-374).

-More detailed analysis of *spaw* and *lefty2* expression is required to address this possibility. Analysis of cell behavior in association with reporter lines for *spaw* or its downstream effectors such as phosphorylated Smad2/3 would help to resolve this question.

3. -Line 292: the cited literature is not on zebrafish.

[RESPONSE]

We have added the following references, which showed that *spaw* expression was initiated near the tailbud/KV, but it did not propagate to the anterior LPM in *spaw* morphants/mutants (line 266-267).

Added references:

Long et al., 2003, Development 130, 2303–2316.

Noël et al., 2013, Nat. Commun. 4, 2754

4. -Line 490: typo

[RESPONSE]

We appreciate the reviewer for pointing this out. We have corrected "The sequence the MOs" to "The sequence of the MOs" (line 499-500).

REVIEWERS' COMMENTS:

Reviewer #1 (Remarks to the Author):

The authors have well addressed my comments and provide important new data in Fig 7 and Supp figures.

I encourage them to add the species in the title. Claiming that their important and insightful observations in the fish can be generalised is overstated without side-by-side comparisons. Their previous work in the chick did not address Nodal directly, nor indirect effect on convergence-extension. The paragraph in the discussion not only states similarities between vertebrate species. Because the fish heart has a very different shape compared to chick and mouse, there are necessarily underlying variations in cell behaviour. So the claim of a general mechanism only brings confusion to my opinion. Advanced quantifications in each species is equally important to reach the conclusion on evolutionary conservation. So that adding fish in the title will not diminish their findings but will clarify the context of their findings.

Reviewer #3 (Remarks to the Author):

The authors have made some substantial modifications to the manuscript which provides more evidence for their model and make it easier to understand their main message. I want to congratulate the authors with their very interesting findings and have no further questions.

Reviewers' comments:

Reviewer #1 (Remarks to the Author):

The authors have well addressed my comments and provide important new data in Fig 7 and Supp figures.

I encourage them to add the species in the title. Claiming that their important and insightful observations in the fish can be generalised is overstated without side-by-side comparisons. Their previous work in the chick did not address Nodal directly, nor indirect effect on convergence-extension. The paragraph in the discussion not only states similarities between vertebrate species. Because the fish heart has a very different shape compared to chick and mouse, there are necessarily underlying variations in cell behaviour. So the claim of a general mechanism only brings confusion to my opinion. Advanced quantifications in each species is equally important to reach the conclusion on evolutionary conservation. So that adding fish in the title will not diminish their findings but will clarify the context of their findings.

[RESPONSE]

We are glad that the reviewer appreciates our new data.

As suggested, we have added the species to the title. After studying the development of the zebrafish, chick, and mouse hearts for many years, we have begun to feel that early development of the fish heart is not as different from that of the amniotes' hearts as it is generally thought. However, as pointed out by the reviewer, from the existing data, it cannot be concluded that the asymmetric cellular behavior that we reported in zebrafish also operates in the asymmetric heart morphogenesis of other species.

Reviewer #3 (Remarks to the Author):

The authors have made some substantial modifications to the manuscript which provides more evidence for their model and make it easier to understand their main message. I want to congratulate the authors with their very interesting findings and have no further questions.

[RESPONSE]

We thank the reviewer for these kind and encouraging comments.